# LIKE OIL AND WATER: GROUP ROBUSTNESS METHODS AND POISONING DEFENSES MAY BE AT ODDS

**Michael-Andrei Panaitescu-Liess** *
University of Maryland
College Park, MD, USA
mpanaite@umd.edu

**Yigitcan Kaya** *
University of California
Santa Barbara, CA, USA
yigitcan@ucsb.edu

**Sicheng Zhu**
University of Maryland
College Park, MD, USA
sczhu@umd.edu

**Furong Huang**
University of Maryland
College Park, MD, USA
furongh@umd.edu

**Tudor Dumitras**
University of Maryland
College Park, MD, USA
tudor@umd.edu

## ABSTRACT

Group robustness has become a major concern in machine learning (ML) as conventional training paradigms were found to produce high error on minority groups. Without explicit group annotations, proposed solutions rely on heuristics that aim to identify and then amplify the minority samples during training. In our work, we first uncover a critical shortcoming of these methods: an inability to distinguish legitimate minority samples from poison samples in the training set. By amplifying poison samples as well, group robustness methods inadvertently boost the success rate of an adversary—e.g., from $0\%$ without amplification to over $97\%$ with it. Notably, we supplement our empirical evidence with an impossibility result proving this inability of a standard heuristic under some assumptions. Moreover, scrutinizing recent poisoning defenses both in centralized and federated learning, we observe that they rely on similar heuristics to identify which samples should be eliminated as poisons. In consequence, minority samples are eliminated along with poisons, which damages group robustness—e.g., from $55\%$ without the removal of the minority samples to $41\%$ with it. Finally, as they pursue opposing goals using similar heuristics, our attempt to alleviate the trade-off by combining group robustness methods and poisoning defenses falls short. By exposing this tension, we also hope to highlight how benchmark-driven ML scholarship can obscure the trade-offs among different metrics with potentially detrimental consequences.

## 1 INTRODUCTION

As ML finds adaptations in many fields with diverse priorities, new metrics of success, aside from prediction performance (e.g., accuracy), have come into play. For example, in security-critical applications, robustness to adversarial examples (Chen et al., 2021) or poisoning attacks (Steinhardt et al., 2017); or, in demographically-sensitive applications, fairness (Hashimoto et al., 2018) or group robustness (Liu et al., 2021) are popular metrics the ML community aims to improve. The sheer number of such metrics has led to a paradigm where researchers demonstrate progress on benchmarks often designed with a single metric in mind.

Recent work has exposed previously unknown trade-offs between some of these metrics, e.g., between privacy and fairness (Bagdasaryan et al., 2019) or between robustness and privacy (Song et al., 2019). As applications in practice require balancing various, often mission-critical, metrics, such unknown trade-offs might have catastrophic consequences. This makes the research into studying the intersection of multiple metrics to identify tensions and interactions particularly crucial. With this motivation, in a systematical quantitative study, our work uncovers an inherent tension between approaches designed for two critical metrics: group robustness methods and poisoning defenses.

---

*Equal contribution.

Group robustness has become a concern as standard training via empirical risk minimization (ERM) has been shown to perform well on an average sample but poorly on samples belonging to under-represented, minority groups (Tatman, 2017). Effective solutions, such as minority upsampling (Byrd & Lipton, 2019), are not always feasible as the explicit group annotations they rely on are often not available due to privacy (e.g., demographic annotations) or financial concerns (e.g., large-scale data sets). To this end, research has proposed heuristics for identifying the minority training samples as a proxy for annotations (Liu et al., 2021). A common observation behind these heuristics is that minority samples are often *difficult to learn* and the model cannot achieve low training error on them. The candidates identified as minority samples are then amplified during training, e.g., through upsampling, which is shown to improve group robustness significantly.

First, we expose a vulnerability (Figure 1): when the training set contains *poison* samples, group robustness heuristics cannot distinguish legitimate minority samples from them. Poison samples are injected by an attacker to teach the model an undesirable behavior, e.g., a backdoor (Saha et al., 2020). As a result, two recent group robustness methods (Sohoni et al., 2020; Liu et al., 2021) end up assisting the attacker by encouraging low error on poison samples along with minority samples—attacker achieves $15 - 97\%$ higher success rate due to amplification. Aiming to understand this vulnerability, we observe that poison samples can be as difficult to learn as minority samples, especially in a realistic attack that can inject only a few samples. This suggests that any group robustness method that relies on difficulty-based heuristics might carry a similar vulnerability. To supplement our empirical results, we prove that under specific assumptions, loss-based difficulty heuristics cannot distinguish between legitimate minority samples and poisons.

Second, on the other side of the coin, we show that poisoning defenses hurt group robustness when the training set contains legitimate minority samples. In particular, we focus on recent sanitization-based defenses in centralized (Yang et al., 2022) and federated learning (Panda et al., 2022) settings. As it is challenging to detect poisons reliably (Shan et al., 2022), these methods pursue a simpler goal by relying on heuristics to identify outliers. Such heuristics are empirically shown to eliminate poisons without hurting the overall accuracy. However, due to providing distinct learning signals during training, we observe that minority samples are often inadvertently identified (and eliminated) as outliers. This poses a trade-off for the defender: the more poisons are eliminated (lower attack success), the more minority samples will be eliminated as well (lower group robustness). In consequence, the accuracy on the minority group drops by up to $15\%$ after applying an effective defense.

Finally, we make an (unsuccessful) attempt to mitigate these tensions by applying poisoning defenses and group robustness methods in tandem. When aiming for low attack success, the defense removes enough minority samples to render group robustness methods ineffective. When aiming for high group robustness, the defense ignores enough poison samples that are still amplified by group robustness, which leads to high attack success. Ultimately, this implies an unintended alignment between defensive and group robustness heuristics. We hope to encourage future work to develop heuristics that conciliate these two critical success metrics.

**In summary, we make the following contributions:** *(i)* We show experimentally that group robustness methods fail to distinguish minority groups from poisons, which leads to the risk of amplifying the attacks, and complement our findings with a theoretical result (Section 4); *(ii)* We find that poisoning defenses also fail to distinguish poisons from under-represented groups and they introduce a risk of lower group robustness in both centralized and federated learning (Section 5 and Appendix A.4); *(iii)* We demonstrate that a straightforward combination of group robustness methods and poisoning defenses cannot fully mitigate these tensions (Appendix A.5) .

## 2 RELATED WORK

**Group Robustness Methods.** Group robustness focuses on training models that obtain good performance on each pre-defined group in the data set. Approaches to group robustness fall into two broad categories. The approaches proposed by Sagawa et al. (2019); Byrd & Lipton (2019); Cao et al. (2019) rely on explicit group annotations during training. For example, group distributionally robust optimization (group DRO) (Sagawa et al., 2019) directly minimizes the worst-group error on the training set. The second set of approaches focuses on a more realistic scenario where annotations are not available during training (Liu et al., 2021; Nam et al., 2020; Sohoni et al., 2020; Namkoong & Duchi, 2017). Our work focuses on a promising type of approach within this set that essentially

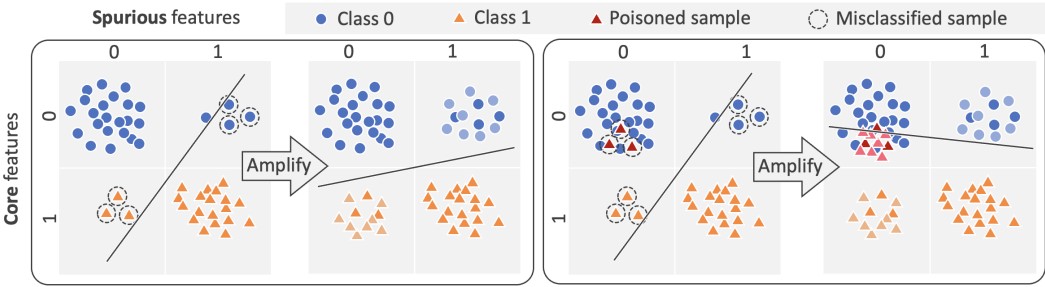

Figure 1: Illustration of group robustness methods without (*left*) and with poisons (*right*) in the training set. For the former, the methods operate regularly as they identify and amplify the minority groups, but for the latter, they also amplify poisons and, therefore, the attacker's influence over the decision boundary. The left and the right panels of each figure correspond respectively to the identification phase of the group robustness methods and the resulting model trained after this phase. Note that the lighter-colored circles and triangles represent amplified points.

aims to obtain pseudo group labels by deploying various heuristics (Liu et al., 2021). We include a discussion on the heuristics used by several other group robustness methods (Zhang et al., 2022; LaBonte et al., 2022; Kim et al., 2022; LaBonte et al., 2024) in Appendix A.2.

**Poisoning Attacks.** In a poisoning attack, the adversary injects a set of malicious samples into the victim's training set. The poisons are designed to induce certain vulnerabilities in the victim model. In dirty-label attacks, the adversary fully controls how the injected sample is crafted and labeled (Gu et al., 2017). In clean-label attacks, however, they can only make minor changes to existing training samples without changing their ground truth label (Suciu et al., 2018). In terms of the attacker's goal, indiscriminate attacks hurt the model's overall accuracy (Koh et al., 2022), targeted attacks cause misclassifications on specific samples (Shafahi et al., 2018), and backdoor attacks teach the model to misclassify any sample that contains a trigger pattern (Gu et al., 2017). In our work, we use a range of poisoning attacks to demonstrate a limitation of group robustness methods in distinguishing legitimate samples from poisons.

**Poisoning Defenses.** Based on their core assumptions, poisoning defenses can be split into three. *(i)* First category corresponds to the assumption that poisons are difficult to be learned and this includes data sanitization defenses that detect anomalies and outliers in the training set (Steinhardt et al., 2017; Chen et al., 2019; Yang et al., 2022). Note that these are popular strategies against poisoning attacks, because of their low computational overhead and minimal impact on accuracy. In this work, we focus on defenses from this category and show that they lead to the problem we identify as they end up eliminating difficult-to-learn minority samples as poisons. *(ii)* Defenses from the second category assume that poisons are easy to be learned, making them suitable against strong adversaries (Li et al., 2021). *(iii)* Third category corresponds to the assumption that poisons follow a different distribution from clean samples. State-of-the-art defenses (Pan et al., 2023; Qi et al., 2023) use a small base set of clean samples to separate the clean training points from the poisons. We discuss the limitations and challenges of using defenses from the second and third categories in Section 6.

## 3 PROBLEM FORMULATION

### 3.1 DEFINITIONS

In this part, we draw inspiration from Sagawa et al. (2019) to define the notion of groups.

**Groups.** Given the set of class labels $\mathcal{Y}$ and spurious attribute (feature) labels $\mathcal{A}$, we consider a data generation process such that $X$ is generated using a class $Y$ and a spurious attribute $A$. We denote a data point by a triplet $(x, y, a)$ and its group (label) by the tuple $(y, a) \in \mathcal{Y} \times \mathcal{A}$. The data distribution of group $(y, a)$ over $\mathcal{X} \times \mathcal{Y} \times \mathcal{A}$ is given by $(D_{(y,a)}, y, a)$. Note that this definition only refers to legitimate and clean-label groups. Later, in Section 4.3, we will also consider a (restricted) definition for poisons as dirty-label samples generated from a legitimate group.

**Minority and Majority Groups.** Since prior work is missing a formal definition for minority and majority groups, we propose one here. Given a dataset $D$ containing samples from the groups, we consider that a group $g = (y, a) \in \mathcal{Y} \times \mathcal{A}$ is a minority group if and only if the number of samples from $g$ is *significantly* lower than the average amount of samples from each group, i.e. $|g| \ll \frac{1}{|\mathcal{Y}| \cdot |\mathcal{A}|} \sum_{h \in \mathcal{Y} \times \mathcal{A}} |h|$, where $|k|$ denotes the number of samples of group $k \in \mathcal{Y} \times \mathcal{A}$ from $D$ and $|\mathcal{Y}|$ and $|\mathcal{A}|$ denote the cardinals of the sets $\mathcal{Y}$ and $\mathcal{A}$, respectively. Also, we consider that a group is a majority group if and only if it is not a minority group.

## 3.2 SETUP

**Datasets.** We consider two benchmark data sets: Waterbirds (Sagawa et al., 2019) and CelebA (Liu et al., 2015). Waterbirds contains $4,795$ training images of "land-bird" and "water-bird" classes, on either land or water backgrounds. CelebA contains of $162,770$ training images of faces, either male or female, and the task proposed by Sagawa et al. (2019) is to classify them as "blond" or "not blond". In our experiments in Section 5, we randomly sample $10\%$ of CelebA. We include experiments with different subset sizes for CelebA in Appendix A.3 and all the other experiments from the paper are performed on the full dataset. We refer to the lowest-represented group (water-birds on land and blond males) as **LRG-1**; and the highest-represented groups (land-birds on land and water-birds on water or blond females, non-blond females and non-blond males) as **HRG**. Additionally, Waterbirds contains a second under-represented group (land-birds on water), which we refer to as **LRG-2**.

**Models.** Following prior work (Liu et al., 2021), we consider standard ResNet architectures (He et al., 2016), starting from ImageNet-pretrained weights. In our main manuscript, we show results on ResNet-18, then we include additional results on ResNet-50 as well as for the scenario when models are trained from scratch in Appendix A.3.

**Group Robustness Methods.** We consider two popular techniques from recent work: Just Train Twice (JTT) (Liu et al., 2021) and GEORGE (Sohoni et al., 2020). Both methods have two main phases: (i) *identification* uses heuristics to identify pseudo group annotations for training samples; and (ii) *amplification* uses these annotations to amplify under-represented groups. In (i), JTT trains a heavily regularized model via ERM and identifies the training samples this model misclassifies as belonging to an under-represented group. In (ii), it simply upsamples these samples to train a second model that achieves higher group robustness. On the other hand, in (i), GEORGE trains a standard model via ERM, clusters its latent representations on the training samples, and treats the cluster labels as pseudo group annotations. In (ii), it applies group DRO (Sagawa et al., 2019) on these groups, which ends up amplifying the samples in smaller clusters as under-represented groups.

**Poisoning Attacks.** We consider (i) a dirty-label backdoor (DLBD) attack that inserts samples containing a trigger with a wrong label; (ii) a sub-population attack (SA) that targets a specific group indiscriminately (Jagielski et al., 2021); and (iii) Gradient Matching (GM) (Geiping et al., 2020), a state-of-the-art clean-label targeted poisoning attack.

Similar to prior work, we consider that $1\%$ of the training set is poisoned for DLBD and GM, and $2\%$ for SA. We also consider other poison percentages in Appendix A.3. For GM, we select the base samples (i.e., the clean training samples the attack modifies into poisons) from LRG-1. For DLBD and SA, we select them from HRG and label them into the same class as LRG-1. For GM, we select 5 target samples from the class that does not contain LRG-1 and launch the attack to force the model to classify them as the class that contains LRG-1. We also consider a setting with 100 target samples and show the results in Appendix A.3. Note that we consider that the attacker is aware of which groups are the easiest to learn so they can craft their poisons accordingly.

**Poisoning Defenses.** In our centralized training experiments, we consider EPIc (Yang et al., 2022), a state-of-the-art technique that iteratively (i) *identifies* the training samples isolated in gradient space as outliers, and (ii) *eliminates* them as potential poisons during training. Additionally, we also consider STRIP (Gao et al., 2019), a run-time backdoor detection defense, and include the results in Appendix A.4. In our federated learning experiments, we consider several robust aggregation mechanisms and poisoning defenses, including coordinate median update (Yin et al., 2018), Trimmed Mean (Yin et al., 2018), and SparseFed (Panda et al., 2022). These techniques aim to sanitize the updates sent by clients and prevent the model from learning outliers.

In Appendix A.1, we provide additional details about our setup and hyper-parameters.

## 3.3 Relevant Metrics

We perform each experiment 3 times and report the average and the standard deviation of its results.

**Accuracy Metrics.** We report two metrics (as percentages) for the prediction performance of a model. First, we report the standard test accuracy (denoted as **ACC**) measured over the entire test set. ACC is dominated by HRG as it does not consider the group labels. To report the group robustness of a model, we measure the Worst-Group Accuracy (denoted as **WGA**). For WGA, we use the ground truth annotations to measure the accuracy on each group (i.e., the percentage of the correctly classified samples from each group). We then report the lowest accuracy among all groups as WGA.

**Identification Success Metrics.** Group robustness methods ideally identify and amplify only the samples in legitimate minority groups, whereas the other groups (including poisons) remain untouched. To report how close we are to the ideal, we measure the Identification-Factor (denoted as **IDNF**) of each ground truth group individually. IDNF measures what percentage of samples in the group end up being amplified. A small gap between IDNF on poisons and IDNF on LRG indicates a failure scenario.

**Attack Success Metrics.** In all attacks, we report Attack Success Rate (denoted as **ASR**) as a percentage. The actual measurement of ASR depends on the attack. For DLBD, we measure the percentage of test samples that are correctly classified in the absence of the trigger, but misclassified in its presence; for GM, the percentage of the misclassified target samples, and for SA, the relative accuracy drop on the target group of the attack over a non-poisoned model. Note that in case a defense not only prevents the accuracy drop by removing the poisons but also slightly increases the accuracy on the targeted group, then the ASR for the SA would be negative.

**Defense Success Metrics.** An ideal defense only eliminates poisons and reduces the ASR, leaving ACC and WGA intact. For EPIc, we measure the Elimination-Factor (denoted as **ELMF**) as the percentage of training samples removed from each individual ground truth group. A large gap between ELMF on LRG and ELMF on HRG indicates disparate impact. For the federated learning defenses (that operate on client updates, not on training samples), we report the drop in WGA and ACC over an undefended model.

## 4 Limitations of Group Robustness Methods

In this section, we show that heuristics deployed by group robustness methods identify the poisons as under-represented and amplify them. We illustrate this vulnerability in Figure 1. We also study the implications of this limitation regarding the ASR of poisoning attacks.

### 4.1 Group Robustness Heuristics Identify Poisons

We start by examining which samples are identified by group robustness methods. For JTT, these are the samples misclassified in the first phase, and, for GEORGE, the samples in the smallest cluster. In Table 1, we present the IDNFs on each group (LRG-1, LRG-2, and HRG) for each method. We do not consider LRG-2 for GEORGE because it creates clusters for each class individually and our poisons are only in the same class as LRG-1. In all experiments, the smallest cluster ends up in the same class as LRG-1, which indicates that GEORGE is working as intended.

Across the board, we see that poisons and LRG-1 samples have the highest IDNF—between $93.4\%$ and $100\%$) Most notably, in many cases, the IDNF on poisons is up to $5\%$ higher than the IDNF on LRG-1. This suggests that group robustness methods are more likely to amplify poisons than legitimate under-represented samples. For Waterbirds, there is an expected gap between the IDNFs on LRG-1 and LRG-2 as LRG-2 contains over $3\times$ more samples than LRG-1, which makes it less difficult to learn. Finally, in all settings, the IDNF on HRG is much lower than the rest—at most $12.5\%$ in the case of GEORGE on Waterbirds.

We also experiment with letting GEORGE automatically adjust the number of clusters for each class by maximizing the silhouette score, as done by Sohoni et al. (2020). This still ends up creating a small cluster that mostly contains the poisons and LRG-1, meaning that it has failed to distinguish between under-represented groups and poisons.

Table 1: The Identification-Factors (IDNFs) of group robustness methods on different groups of samples in the training set. We highlight the alarming cases where the poison samples are more amplified than the lowest-represented group.

| METHOD | DATASET | ATTACK | POISONS | LRG-1 | LRG-2 | HRG |
|--------|---------|--------|---------|-------|-------|-----|
| JTT | WATERBIRDS | DLBD | $\textbf{98.5} \pm 1.2\%$ | $94.6 \pm 1.4\%$ | $36.9 \pm 1.8\%$ | $5.0 \pm 0.3\%$ |
| JTT | WATERBIRDS | SA | $\textbf{98.2} \pm 0.6\%$ | $94.6 \pm 1.7\%$ | $38.7 \pm 3.1\%$ | $4.8 \pm 0.3\%$ |
| JTT | WATERBIRDS | GM | $\textbf{100.0} \pm 0.0\%$ | $96.2 \pm 6.4\%$ | $35.3 \pm 1.9\%$ | $5.4 \pm 0.3\%$ |
| JTT | CELEBA | DLBD | $\textbf{99.9} \pm 0.0\%$ | $96.7 \pm 0.2\%$ | $N/A$ | $9.8 \pm 0.5\%$ |
| GEORGE | WATERBIRDS | DLBD | $\textbf{98.5} \pm 1.2\%$ | $93.4 \pm 1.0\%$ | $-$ | $12.5 \pm 1.2\%$ |

**Takeaways.** The heuristics used by group robustness methods achieve a high recall in identifying LRG. However, they often identify most of the poisons as a minority group, too—between $98.2\%$ and $100\%$—which even exceeds the recall on the legitimate LRG. Overall, this supports our claim that current group robustness methods are limited in distinguishing between minority groups and poisons.

## 4.2 GROUP ROBUSTNESS METHODS AMPLIFY POISONS

After finding that group robustness methods identify poisons as an under-represented group, in this section, we study how this impacts poisoning attacks and their success. To this end, we consider three evaluation settings. In the *standard* case, we apply the group robustness method as-is. In the *ideal* and *worst* cases, we intervene in the method to prevent it from amplifying any poison or to force it to amplify all poisons, respectively. These interventions aim to isolate the impact of amplifying poisons on the attack success rate (ASR). We implement these interventions by manually removing all poisons from (ideal) or placing all poisons into (worst) the set of samples identified by group robustness methods.

We present the results in Table 2, across different attacks, methods, and datasets. We first note that the ASR gap between the worst case and the standard case is often minimal. This highlights that the boost the attacker gains due to the shortcomings of group robustness heuristics is as significant as it can get. The large gap between $(6.7\% - 97.4\%)$ the standard and ideal cases shows an opportunity for better heuristics. We believe the larger amplification in the case of CelebA stems from the fact that this is a more complex data set with more variability and samples, which makes it easier for poisoning. Note that most of the models maintain a relatively high WGA (at least $74.1\%$), which shows that group robustness methods are working as intended, but still worse than the case without any poisoning ($86.7\%$ as in Liu et al. (2021)). The only exception to this is the case of SA, where the WGA drops to $60.9\%$. However, this is expected as the goal of this attack is to hurt the accuracy on a specific group. Finally, the models tend to maintain a fairly high standard ACC, except against SA as it is an indiscriminate attack, which provides a sanity check to our results.

Additionally, in Appendix A.3, we study the impact of the hyper-parameters (early stopping for the identification model and upsampling factor) and consider more settings (different amount of poisoned samples, more targets for GM attack and using a larger model, as well as training the models from scratch). We observe that the results are consistent with our previous findings.

**Takeaways.** Due to identifying poisons as an under-represented group, group robustness methods end up directly or indirectly amplifying them. We show that this leads to a boost in the success rate of poisoning attacks and, generally, this boost is almost as high as in the worst-case scenario.

## 4.3 IMPOSSIBILITY RESULT

In this section, we show that under some assumptions, loss-based identification methods are inherently ineffective in separating legitimate minority groups from poisons.

We consider a binary classification problem with one binary spurious attribute ($\mathcal{Y} = \mathcal{A} = \{0, 1\}$). We denote the identification model (i.e., the model used in the identification phase of the group robustness methods) by $I : \mathcal{X} \rightarrow [0, 1]^{|\mathcal{Y}|}$ and its output probability corresponding to the $i^{th}$ class by $I(x)_i$ (i.e., the softmax output on class $i$). For a group $(y, a) \in \mathcal{Y} \times \mathcal{A}$, we denote

Table 2: Evaluating the impact of amplification in group robustness methods on worst group accuracy (WGA), attack success rate (ASR), and test accuracy (ACC). We highlight when there is a small gap in ASR between the worst and standard cases as this indicates that a method has given an advantage to the adversary.

| METHOD | DATASET | ATTACK | CASE | WGA | ASR | ACC |
|---|---|---|---|---|---|---|
| JTT | WATERBIRDS | DLBD | WORST | $76.9 \pm 3.5\%$ | $\mathbf{20.9 \pm 7.9\%}$ | $86.4 \pm 1.2\%$ |
| | | | STANDARD | $78.0 \pm 4.1\%$ | $\mathbf{20.4 \pm 5.9\%}$ | $86.7 \pm 1.2\%$ |
| | | | IDEAL | $81.4 \pm 0.9\%$ | $\mathbf{0.5 \pm 0.3\%}$ | $91.0 \pm 0.4\%$ |
| JTT | WATERBIRDS | SA | WORST | $60.9 \pm 4.5\%$ | $\mathbf{24.0 \pm 3.3\%}$ | $70.9 \pm 2.3\%$ |
| | | | STANDARD | $61.6 \pm 6.8\%$ | $\mathbf{31.4 \pm 6.0\%}$ | $66.0 \pm 3.8\%$ |
| | | | IDEAL | $82.3 \pm 1.5\%$ | $\mathbf{0.8 \pm 1.3\%}$ | $90.8 \pm 0.3\%$ |
| JTT | WATERBIRDS | GM | WORST | $76.1 \pm 3.2\%$ | $\mathbf{20.0 \pm 0.0\%}$ | $89.9 \pm 1.0\%$ |
| | | | STANDARD | $76.1 \pm 3.2\%$ | $\mathbf{20.0 \pm 0.0\%}$ | $89.9 \pm 1.0\%$ |
| | | | IDEAL | $75.9 \pm 0.8\%$ | $\mathbf{13.3 \pm 11.5\%}$ | $91.3 \pm 0.1\%$ |
| JTT | CELEBA | DLBD | WORST | $79.7 \pm 0.9\%$ | $\mathbf{97.7 \pm 0.1\%}$ | $82.9 \pm 0.8\%$ |
| | | | STANDARD | $79.3 \pm 0.2\%$ | $\mathbf{97.7 \pm 0.1\%}$ | $82.6 \pm 0.2\%$ |
| | | | IDEAL | $79.2 \pm 1.1\%$ | $\mathbf{0.3 \pm 0.0\%}$ | $83.6 \pm 1.4\%$ |
| GEORGE | WATERBIRDS | DLBD | WORST | $74.1 \pm 1.7\%$ | $\mathbf{16.5 \pm 2.9\%}$ | $76.4 \pm 3.4\%$ |
| | | | STANDARD | $77.3 \pm 2.5\%$ | $\mathbf{15.8 \pm 3.9\%}$ | $79.4 \pm 1.6\%$ |
| | | | IDEAL | $79.3 \pm 2.1\%$ | $\mathbf{0.4 \pm 0.0\%}$ | $93.1 \pm 1.4\%$ |

$p_{y,a} := \mathbb{E}_{(x,y,a)\sim(D_{(y,a)},y,a)}[I(x)_y]$ (i.e., the expected class $y$ probability output of the identification model on a sample from the group $(y,a)$). In the rest of the paper we will refer to this as *expected class probability*. We consider all the groups (minority and majority) to be clean-label and poisons to be dirty-label. To generate poison samples starting from legitimate samples from group $(y,a)$, the attacker needs to flip their label (this corresponds to the FeatureMatch and Label Flipping setting of the Subpopulation Attack from Jagielski et al. (2021)), so the distribution for the poisons $(y,a)^*$ is $(D_{(y,a)}, 1-y, a)$ and we define $p_{y,a}^* := \mathbb{E}_{(x,1-y,a)\sim(D_{(y,a)},1-y,a)}[I(x)_{1-y}]$. We denote the maximum expected class probability on a group as $p_{y_m,a_m} := max_{y,a\in\{0,1\}}p_{y,a}$. Note that as the number of dirty-label samples is generally much lower than the number of clean samples (a realistic attacker cannot insert a lot of poisons), we can assume that $p_{y_m,a_m} > max_{y,a\in\{0,1\}}p_{y,a}^*$.

**Lemma 4.1.** *For the setting described above, if we assume that there are no ties in maximum expected class probability among groups, then the identification model has less expected class probability on the poisons $(y_m,a_m)^*$ in comparison to any legitimate group.*

Proof. We include the proof in Appendix A.6.

**Theorem 4.2.** *We consider the same setting as in Lemma 4.1. We denote the poisons $(y_m,a_m)^*$ by $g_p$ and let $(y,a) := g_c$ be any group of samples (e.g., a legitimate minority group). Also, we denote $I(x)_{1-y_m}$ for $(x,1-y_m,a_m) \sim (D_{g_p},1-y_m,a_m)$ by $G_p$ and $I(x)_y$ for $(x,y,a) \sim (D_{g_c},y,a)$ by $G_c$ and the cross-entropy loss on $G \in \{G_c,G_p\}$ by $L(G)$. We assume $G_p$ and $G_c$ are independent and $Var(G_p) = Var(G_c) := \sigma^2$ (i.e., the variances of the class probability for the identification model are equal for the legitimate group and for the poisons) and denote $\mathbb{E}(G_p) := \mu_p$ and $\mathbb{E}(G_c) := \mu_c$. Then, for any $\epsilon \in (0,1)$, if $\sigma \leqslant \sqrt{\frac{1}{\sqrt{1-\epsilon}}-1}\cdot\frac{\mu_c-\mu_p}{2}$, we have $\mathbb{P}(L(G_c) < L(G_p)) > 1-\epsilon$.*

Proof. We include the proof in Appendix A.6.

**Takeaways.** An identification model amplifies hard-to-learn samples as they are hypothesized to approximate legitimate minority groups. Our result in Theorem 4.2 undermines this common hypothesis by proving that, under some conditions, the poisons are harder to learn than minority group samples and, therefore, loss-based thresholding will fail to separate them with a high probability. The strong correlations among popular sample difficulty metrics (Wu et al., 2020), such as the loss value or learning epoch, give our result a broader applicability despite our theorem's focus on a prototypical loss-based thresholding scheme.

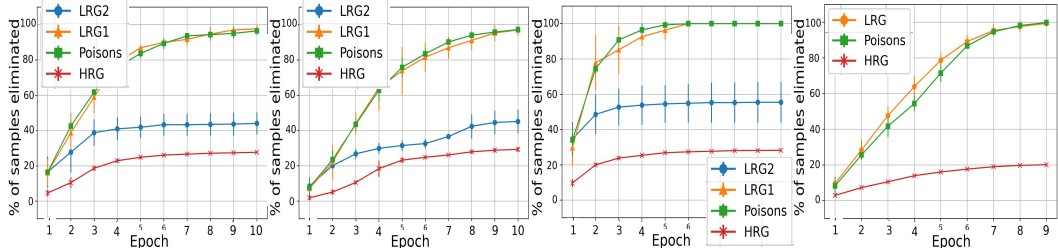

Figure 2: The elimination disparity between the under-represented (LRG) and over-represented (HRG) groups against EPIc. The x-axes show the iterations of EPIc, and y-axes show the Elimination-Factor (ELMF) for each group. From *left* to *right*, the first three plots are for DLBD, SA, GM attacks on Waterbirds and the last one is for DLBD on CelebA.

Table 3: The impact of EPIc on group robustness, measured by the Worst-Group Accuracy (WGA).

| DATASET | ATTACK | CASE | WGA | ASR | ACC |
|---------|--------|------|-----|-----|-----|
| WATERBIRDS | DLBD | IDEAL | $\mathbf{59.0} \pm 2.5\%$ | $0.1 \pm 0.1\%$ | $95.5 \pm 0.1\%$ |
| | | STANDARD | $\mathbf{55.8} \pm 6.0\%$ | $0.1 \pm 0.0\%$ | $95.0 \pm 0.1\%$ |
| | | WORST | $\mathbf{50.7} \pm 6.9\%$ | $0.1 \pm 0.0\%$ | $94.3 \pm 0.8\%$ |
| WATERBIRDS | SA | IDEAL | $\mathbf{59.1} \pm 3.0\%$ | $-0.3 \pm 1.8\%$ | $94.6 \pm 0.4\%$ |
| | | STANDARD | $\mathbf{55.3} \pm 3.2\%$ | $0.2 \pm 1.7\%$ | $94.2 \pm 0.4\%$ |
| | | WORST | $\mathbf{48.1} \pm 7.3\%$ | $-0.1 \pm 2.3\%$ | $93.9 \pm 0.3\%$ |
| WATERBIRDS | GM | IDEAL | $\mathbf{50.2} \pm 2.0\%$ | $13.3 \pm 11.5\%$ | $95.8 \pm 0.3\%$ |
| | | STANDARD | $\mathbf{43.5} \pm 7.8\%$ | $13.3 \pm 11.5\%$ | $94.6 \pm 0.5\%$ |
| | | WORST | $\mathbf{44.3} \pm 8.3\%$ | $13.3 \pm 11.5\%$ | $94.6 \pm 0.4\%$ |
| CELEBA* | DLBD | IDEAL | $\mathbf{54.8} \pm 3.2\%$ | $0.3 \pm 0.2\%$ | $93.9 \pm 0.0\%$ |
| | | STANDARD | $\mathbf{40.5} \pm 4.5\%$ | $0.1 \pm 0.0\%$ | $94.0 \pm 0.1\%$ |
| | | WORST | $\mathbf{34.8} \pm 1.1\%$ | $0.0 \pm 0.0\%$ | $93.7 \pm 0.3\%$ |

## 5 POISONING DEFENSES HAVE DISPARATE IMPACT

After establishing how group robustness methods amplify poisons, in this section, we investigate whether poisoning defenses have any undesirable impact on under-represented samples and group robustness.

### 5.1 POISONING DEFENSES ELIMINATE MINORITY SAMPLES

We start our investigation by studying the impact of EPIc on under-represented samples. In Figure 2, we show the Elimination-Factor (ELMF) of EPIc in four different settings. We observe that generally the poisons and LRG-1 samples are eliminated at a similar rate (the two curves are close to each other, across the board). On the other hand, the ELMF on HRG samples is always significantly less. Interestingly, ELMF on LRG-2 samples from Waterbirds is less than the poisons and LRG-1 samples and more than HRG samples. A large (reaching almost $100\%$) ELMF on poisons shows that EPIc is indeed an effective poisoning defense. However, by eliminating most poisons, EPIc also eliminates most LRG-1 samples as well. This suggests that legitimate under-represented samples are strong outliers from the perspective of EPIc, which demonstrates a disparate impact.

### 5.2 POISONING DEFENSES REDUCE GROUP ROBUSTNESS

Here, we study the effect of EPIc on group robustness by considering three scenarios. In ideal and worst-case scenarios, we make interventions on EPIc to never eliminate any under-represented sample or to eliminate under-represented samples as early as possible, respectively. In standard EPIc, we make no intervention and apply the defense as-is. Through interventions, we hope to isolate the impact of EPIc on minority samples. For each scenario, we report WGA to measure group robustness.

We present the results in Table 3. First, we observe that in different datasets and attacks, EPIc reduces the WGA by $3.2\% - 14.3\%$. The most damage happens on CelebA data set as, we believe, it is more complex than Waterbirds. Overall, the ASR is low, indicating that EPIc works properly, with one exception for the GM attack, where the ASR is $13.3\%$ for all three scenarios. In all cases, ACC is high, relatively close to ACC reported in prior work (Liu et al., 2021). We see a significant gap in WGA after applying EPIc and applying group robustness methods (in Table 2), almost up to $40\%$. This is expected as EPIc does not make any attempts to preserve group robustness. In Appendix A.5, we make an effort towards applying poisoning defenses while preserving group robustness.

Additionally, we study the impact that robust aggregation mechanisms and poisoning defenses have on the under-represented groups in Federated Learning, as well as more settings including different amounts of poisoned samples. We show the results in Appendix A.4 and observe that they are consistent with our previous findings.

**Takeaways.** Poisoning defenses either aim to identify and remove the outliers or make it more difficult for the model to learn poisons (e.g., in Federated Learning). They, however, also end up making minority groups more difficult to learn as well, which hurts the group robustness of the trained model. This shows that, despite the common practice, ACC can be over-optimistic in gauging the impact of a poisoning defense in the presence of under-represented groups.

## 6    DISCUSSION AND FUTURE WORK

In this work, we have focused only on defenses from the first category in Section 2 (that consider poisons as difficult to learn). We have exposed their vulnerability and the potentially harmful consequences of these defenses. We believe that defenses from the second category (that consider poisons as easy to learn), would not lead to these problems. However, it is important to note that, poisons will not be easy to learn in realistic attacks where adversaries can only inject a limited number of poisons, violating the assumption. As a result, defenses from this category would be ineffective against such attacks and, therefore, group robustness methods would still inadvertently offer a needed boost to the weaker adversary. Finally, for the third category of defenses (poisons are different from clean samples), the state-of-the-art defenses (Pan et al., 2023; Qi et al., 2023) use a small base set (10-1000 samples) to model the distribution of clean samples and identify the training points most distinct from this distribution as poisons. These base sets are often assumed to follow the same distribution as the clean training set, which makes them unlikely to contain sufficiently many minority samples. We hypothesize that this will cause such defenses to still eliminate clean minority samples as poisons and hurt the WGA. However, it might be possible to avoid this problem by providing such defenses with carefully curated base sets that are balanced (in terms of groups) and free from poisons. In this case, instead of mistakenly penalizing difficult clean samples as poisons, they can isolate the real poisons from a more inclusive clean distribution captured by the base set. However, research suggests that making a base set poison-free (Zeng et al., 2022) or collecting enough labeled minority samples that capture their distribution properly (Lokhande et al., 2022) might be difficult in practice. We believe addressing these challenges is a promising avenue for future work to reconcile between group robustness and poisoning resilience.

## 7    CONCLUSIONS

In this work, we demonstrate a significant tension involving two critical metrics studied in the ML community: group robustness and poisoning resilience. The objective of group robustness methods is to amplify minority groups in the training set and create more equitable models. Our findings reveal that the samples injected by poisoning attacks consistently mislead these methods into amplifying them, resulting in an undesirable boost to the adversary. On the other hand, poisoning defenses aim to prevent attacks by removing problematic samples from the training set. However, these defenses remove legitimate under-represented samples as well, hence compromising the model's equity. Finally, we wish to emphasize the pressing need for the ML community to focus on the development of new methods that tackle the inherent challenges posed by data poisoning attacks and group robustness methods.

ACKNOWLEDGMENTS

Panaitescu-Liess, Zhu and Huang are supported by National Science Foundation NSF-IIS-2147276 FAI, DOD-ONR-Office of Naval Research under award number N00014-22-1-2335, DOD-AFOSR-Air Force Office of Scientific Research under award number FA9550-23-1-0048, DOD-DARPA-Defense Advanced Research Projects Agency Guaranteeing AI Robustness against Deception (GARD) HR00112020007, Adobe, Capital One and JP Morgan faculty fellowships.

Kaya is supported by the Intelligence Community Postdoctoral Fellowship.

This effort was partially supported by the Intelligence Advanced Research Projects Agency (IARPA) under the contract W911NF20C0035. The content of this paper does not necessarily reflect the position or the policy of the Government, and no official endorsement should be inferred.

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

Table 4: Proportions of each group in Waterbirds dataset. Labels 0 and 1 correspond to the classes landbirds and waterbirds, respectively. Attributes 0 and 1 correspond to the spurious features land and water, respectively.

| Group | Gr. 0 | Gr. 1 | Gr. 2 | Gr. 3 | Poisons |
|---|---|---|---|---|---|
| Label / Attribute | 0/0 | 0/1 | 1/0 | 1/1 | 1/0 |
| Proportion | 72.0% | 3.8% | 1.2% | 22.0% | 1.0% |

# A  Supplementary Material

## A.1  Additional details on the experimental setup

For the models trained with JTT on Waterbirds, unless differently specified, we use SGD with momentum with a factor of $0.9$ as the optimizer, a batch size of $128$, learning rate of $1e-5$ and weight decay of $1.0$. We stop the identification model after $60$ epochs, use an upsampling factor of $100$, train the final model up to $100$ epochs and choose the model that has the best WGA on the validation dataset, then report the results on the test dataset. In case of CelebA, we use the same hyper-parameters as for Waterbirds, with a few exceptions: weight decay of $0.1$, early stopping for the identification model after $1$ epoch and we use an upsampling factor of $50$. Also, we train the final model up to $5$ epochs and consider the same procedure to choose the best model as above. We build our experiments on top of the official code from [1].

For the models trained with GEORGE, we use a similar optimizer, batch size, learning rate, weight decay as in the Waterbirds case from above. For clustering we let the model find the optimal number of clusters (up to 10) for each class based on Silhouette criterion as in Sohoni et al. (2020). We train the final model up to $300$ epochs and use the official code from [2]

For DLBD, the trigger is a 25x25 white square placed 1 pixel away from the bottom-right corner of the poisoned samples.

For SA, we consider FeatureMatch and Label Flipping to create poisoned samples as in prior work (Jagielski et al., 2021).

For GM, we use the multi-target version of the attack, an epsilon of $16$ and $8$ restarts. To build the poisoned samples, we rely on the official code from [3].

For EPIc, in case of Waterbirds, we consider SGD with momentum with a factor of $0.9$ as the optimizer, a batch size of $128$, learning rate of $1e-2$ and weight decay of $1e-4$. We train the models up to $40$ epochs and consider the same selection criterion for the model based on WGA on the validation set as above. In the case of CelebA, we change the learning rate to $1e-3$ and we train the models up to $10$ epochs. We build our experiments on top of the official code from [4].

For the federated learning experiments, we built the implementation on top of the code from here [5]. We consider a total of $100$ users and $10\%$ of them are chosen at each round. We use SGD with momentum with a factor of $0.9$ as the optimizer, a local batch size of $128$ and learning rate of $1e-2$. Also, we train each local model for $10$ epochs at each round. In case of the non-IID setting, we consider that only $10\%$ of the users have samples from the under-represented groups.

For the federated learning defenses, in the case of Trimmed Mean, we remove the lowest and highest two values for each coordinate in the update and in the case of SparseFed we use a value of $400,000$ for the number of parameters that we keep at each step which is equivalent to keeping less than $5\%$ of the parameters. We consider a momentum factor of $\rho = 0.9$ as in prior work (Panda et al., 2022) or not using momentum ($\rho = 0$) in our experiments.

---

[1] https://github.com/anniesch/jtt
[2] https://github.com/HazyResearch/hidden-stratification
[3] https://github.com/JonasGeiping/poisoning-gradient-matching
[4] https://github.com/YuYang0901/EPIC
[5] https://github.com/AshwinRJ/Federated-Learning-PyTorch

Table 5: Proportions of each group in 10% of CelebA and Full CelebA. Labels 0 and 1 correspond to the classes non-blond and blond, respectively. Attributes 0 and 1 correspond to the spurious features female and male, respectively.

| Group | Gr. 0 | Gr. 1 | Gr. 2 | Gr. 3 | Poisons |
|---|---|---|---|---|---|
| Label / Attribute | 0/0 | 0/1 | 1/0 | 1/1 | 1/1 |
| Proportion for 10% CelebA | 44.7% | 39.6% | 13.8% | 0.9% | 1.0% |
| Proportion for Full CelebA | 44.0% | 40.1% | 14.1% | 0.9% | 1.0% |

In Tables 4 and 5, we show the proportions of each group in Waterbirds and CelebA datasets, respectively.

## A.2    Discussion on Heuristics used by Group Robustness Methods

Although we experimented with methods that specifically use a loss-based heuristic, we believe that many other group robustness methods rely on similar heuristics, perhaps less directly. In this section, we discuss four methods from recent literature that implemented seemingly different heuristics that will align with loss-based heuristics.

Zhang et al. (2022) applies contrastive learning to push together the representations of training samples that are labeled into the same class but predicted differently. Since, the low-budget poisoning attacks we consider often generate hard-to-learn, misclassified samples, we believe this method will also suffer from the tension we identify, e.g., misclassified poison samples of class will be represented similarly to correctly classified samples of class, which will amplify their effectiveness as the model becomes less likely to misclassify them.

LaBonte et al. (2022) uses the disagreements between different forward passes of the model with dropout to find which training samples to amplify. Different dropout forward passes are known to disagree on samples where the model has high uncertainty (Gal & Ghahramani, 2016), which are often hard-to-learn, higher-loss samples. As a result, we believe this method will also suffer from the same tension we identified, e.g., the hard-to-learn poison samples we generated will create dropout disagreements and will be amplified.

Kim et al. (2022) relies on disagreements among the members of an ensemble of models to identify the "biased" samples (i.e., the samples that contain spurious correlations). If a sample has low ensemble agreement, it will be amplified by this method. Ensemble disagreement is a known uncertainty metric (Beluch et al., 2018) in the literature. As a result, we believe that this method will also suffer from the same tension we identified, e.g., the hard-to-learn poison samples will be misclassified by more members of the ensemble, and will be amplified.

LaBonte et al. (2024) constructs a reweighting set based on either (1) ERM model's misclassifications, or (2) the misclassifications of an early-stopping model, or (3) dropout disagreement (similar to the Dropout Disagreement paper) or (4) the disagreements between the ERM and early-stopping models. All these heuristics to identify minority group samples will end up identifying the hard-to-learn poisoning samples as well. The authors identified (4) as the most promising heuristic. It is known that hard-to-classify samples are learned at later iterations of model training (Baldock et al., 2021). As a result, we believe that method will also suffer from the same tension we identified, e.g., the hard-to-learn poison samples will be classified differently by the early-stopping and regular models, and will be amplified.

These common threads show that, despite not using the loss heuristic directly, many methods in this line of work rely on related ideas that will amplify uncertain, misclassified, high-loss, or hard-to-learn samples. Moreover, it is known that many example difficulty metrics are highly correlated with one another (Baldock et al., 2021; Carlini et al., 2019). Ultimately, we believe that these related heuristics cannot avoid the tension we identified because low-budget poisoning attacks generate hard-to-learn samples.

### A.3 MORE RESULTS ON LIMITATIONS OF THE GROUP ROBUSTNESS METHODS

**Impact of the Hyper-parameters.** Here, we study how the hyper-parameters of JTT impact our findings from Section 2. We consider two hyper-parameters: *early stopping* for the model in the first phase used to identify LRG; and the *upsampling factor* for the samples identified samples. Our previous experiments use the original hyper-parameters, i.e., early stopping is set to 60 epochs, and upsampling factor is set to $100\times$. However, due to the introduction of poisoning, these hyper-parameters might not be optimal anymore. To this end, we search for new hyper-parameters in a grid (early stopping $\in \{40, 80, 100, 200\}$ and upsampling factors $\in \{20, 50, 150\}$) against DLBD attack on Waterbirds.

We present the results in Tables 6 and 7. We observe that the results are consistent with our previous findings: ASRs are very close between the standard and the worst cases (i.e., high amplification), while both WGA and ACC are still relatively high with the exceptions when early stopping is set to 40 and when the upsampling factor is set to 150. In these cases, the WGAs are only $65.3\%$ and $64.6\%$, respectively. For the former, we believe that it could be due to how earlier stopping makes identification less selective (lower precision) and ends up upsampling HRG as well. Also, the ASR is higher by more than $15\%$ compared to the other early stopping values. For the latter, we believe that it is because the amplified samples appear too often during training compared to the non-amplified ones. Also, surprisingly, using smaller upsampling factors of $50\times$ or $20\times$ instead of $100\times$ makes the attack even more powerful by $8.4\%$ and $46.2\%$ respectively, in terms of ASR.

**Additional Settings.** For completeness, in this section, we analyze several other scenarios on Waterbirds data set, including using different poisons percentages, different numbers of targets for the GM attack, different model architectures and different base samples. In Table 8, we consider JTT and GEORGE with the DLBD attack using several poison percentages in the range $0.4\% - 2\%$. The results are consistent with our previous findings. Additionally, we observe that an attacker who could introduce $2\%$ poisoned samples, generally obtains a higher ASR (of $52.5\% - 59.8\%$). In Table 9, we compare the results in two settings for the GM attack: (i) when the attack has 5 targeted samples (same as our previous experiments) and (ii) when the attack has 100 targeted samples. The results show that, in the case of 100 targets, the amplification is as high as it could be. The only distinction from the case with 5 targets is an overall lower ASR, but this is expected as the attack becomes more difficult as it targets more samples. We also consider a larger architecture, ResNet-50, instead of ResNet-18, for the DLBD attack and several scenarios, including training the models from scratch. Our results in Table 10 are generally consistent, however, the ASR is significantly higher against ResNet-50. We believe the additional learning capacity of ResNet-50 over ResNet-18 facilitates the attack. Also, we observe that training the models from scratch and tuning them for high group robustness (WGA) damages the overall accuracy (ACC) significantly. This challenge explains the prior practice of using pre-trained models as they can alleviate this trade-off to some extent. In Table 11, we consider base samples from different groups of CelebA for DLBD. We observe that the easier a group is to learn (i.e., less average loss), the higher the ASR will be if the attacker crafts a DLBD attack with base samples from that group. In Table 12, we consider the scenario when the attacker is not aware of the presence of groups, so they just attack one of the two classes of CelebA, with the DLBD. We observe high amplification (i.e., the small gap between the worst and standard cases) no matter which class is targeted by the attacker. As a result, even a less informed attacker, who is unaware of the groups, would benefit from the amplification. In Table 13 we consider the Subpopulation Attack (SA) on CelebA when applying JTT and without applying any group robustness method (ERM) as a baseline. We observe that the amplification is generally high (large ASR gap between JTT and the baseline) while JTT still works as intended as it improves the WGA over the baseline. Additionally, in Table 14, we consider AFR (Qiu et al., 2023) on MultiNLI dataset (Williams et al., 2018) and observe similar results. We also consider several datasets from Spawrious benchmark (Lynch et al., 2023). We show the percentage of samples amplified by JTT from each group in Table 15. Again, the results are consistent with our previous findings. To conclude, our extensive experiments show that the limitation of group robustness methods is consistent across different settings, suggesting that this might be an inherent vulnerability.

### A.4 MORE RESULTS ON THE LIMITATIONS OF THE POISONING DEFENSES

**Federated Learning Scenarios.** To demonstrate that the limitation of EPIc we identified (in Section 5) is consistent in other defenses, we also run experiments on federated learning, where

Table 6: Evaluating the impact of amplification in JTT on worst group accuracy (WGA), attack success rate (ASR) and test accuracy (ACC) when considering several early stopping epochs for the identification model.

| EARLY STOPPING | CASE | WGA | ASR | ACC |
|---|---|---|---|---|
| 40 | WORST | $65.4 \pm 9.4\%$ | $\mathbf{46.3} \pm 30.6\%$ | $83.3 \pm 4.4\%$ |
| | STANDARD | $65.3 \pm 9.3\%$ | $\mathbf{45.7} \pm 31.5\%$ | $83.2 \pm 4.3\%$ |
| | IDEAL | $76.8 \pm 3.2\%$ | $\mathbf{0.4} \pm 0.1\%$ | $91.5 \pm 0.5\%$ |
| 60 | WORST | $76.9 \pm 3.5\%$ | $\mathbf{20.9} \pm 7.9\%$ | $86.4 \pm 1.2\%$ |
| | STANDARD | $78.0 \pm 4.1\%$ | $\mathbf{20.4} \pm 5.9\%$ | $86.7 \pm 1.2\%$ |
| | IDEAL | $81.4 \pm 0.9\%$ | $\mathbf{0.5} \pm 0.3\%$ | $91.0 \pm 0.4\%$ |
| 80 | WORST | $80.3 \pm 2.5\%$ | $\mathbf{18.7} \pm 5.9\%$ | $87.3 \pm 0.8\%$ |
| | STANDARD | $80.9 \pm 2.8\%$ | $\mathbf{17.7} \pm 5.5\%$ | $87.6 \pm 1.0\%$ |
| | IDEAL | $82.2 \pm 2.4\%$ | $\mathbf{0.5} \pm 0.1\%$ | $91.3 \pm 0.5\%$ |
| 100 | WORST | $81.8 \pm 2.0\%$ | $\mathbf{19.7} \pm 8.9\%$ | $87.6 \pm 1.2\%$ |
| | STANDARD | $82.0 \pm 2.1\%$ | $\mathbf{23.4} \pm 8.7\%$ | $87.2 \pm 0.9\%$ |
| | IDEAL | $83.3 \pm 0.9\%$ | $\mathbf{0.9} \pm 0.1\%$ | $91.0 \pm 0.4\%$ |
| 200 | WORST | $83.5 \pm 2.4\%$ | $\mathbf{27.8} \pm 9.5\%$ | $87.9 \pm 1.7\%$ |
| | STANDARD | $83.1 \pm 2.1\%$ | $\mathbf{29.5} \pm 8.5\%$ | $87.5 \pm 1.3\%$ |
| | IDEAL | $84.5 \pm 1.5\%$ | $\mathbf{0.7} \pm 0.1\%$ | $90.0 \pm 1.7\%$ |

Table 7: Evaluating the impact of amplification in JTT on worst group accuracy (WGA), attack success rate (ASR) and test accuracy (ACC) when considering several upsampling factors.

| UPSAMPLING FACTOR | CASE | WGA | ASR | ACC |
|---|---|---|---|---|
| 20 | WORST | $71.9 \pm 4.0\%$ | $\mathbf{69.5} \pm 9.1\%$ | $91.4 \pm 0.6\%$ |
| | STANDARD | $72.3 \pm 4.5\%$ | $\mathbf{66.6} \pm 10.0\%$ | $91.5 \pm 0.6\%$ |
| | IDEAL | $74.9 \pm 2.3\%$ | $\mathbf{1.3} \pm 0.3\%$ | $93.0 \pm 0.3\%$ |
| 50 | WORST | $79.7 \pm 1.8\%$ | $\mathbf{33.1} \pm 3.5\%$ | $91.4 \pm 0.5\%$ |
| | STANDARD | $79.6 \pm 1.9\%$ | $\mathbf{28.8} \pm 6.6\%$ | $91.9 \pm 0.6\%$ |
| | IDEAL | $79.5 \pm 1.0\%$ | $\mathbf{0.9} \pm 0.4\%$ | $92.6 \pm 0.4\%$ |
| 100 | WORST | $76.9 \pm 3.5\%$ | $\mathbf{20.9} \pm 7.9\%$ | $86.4 \pm 1.2\%$ |
| | STANDARD | $78.0 \pm 4.1\%$ | $\mathbf{20.4} \pm 5.9\%$ | $86.7 \pm 1.2\%$ |
| | IDEAL | $81.4 \pm 0.9\%$ | $\mathbf{0.5} \pm 0.3\%$ | $91.0 \pm 0.4\%$ |
| 150 | WORST | $65.7 \pm 10.9\%$ | $\mathbf{29.2} \pm 29.5\%$ | $74.1 \pm 3.7\%$ |
| | STANDARD | $64.6 \pm 9.9\%$ | $\mathbf{29.4} \pm 34.4\%$ | $73.2 \pm 3.8\%$ |
| | IDEAL | $73.0 \pm 13.2\%$ | $\mathbf{0.7} \pm 0.1\%$ | $84.3 \pm 7.1\%$ |

Table 8: Evaluating the impact of amplification in group robustness methods on worst group accuracy (WGA), attack success rate (ASR) and test accuracy (ACC) when considering several poison percentages for the DLBD attack.

| POISONS | METHOD | CASE | WGA | ASR | ACC |
|---|---|---|---|---|---|
| 0.4% | JTT | WORST | $78.7 \pm 2.0\%$ | $\mathbf{4.8} \pm 0.7\%$ | $89.3 \pm 1.2\%$ |
| | | STANDARD | $78.6 \pm 2.0\%$ | $\mathbf{3.5} \pm 1.7\%$ | $89.1 \pm 1.0\%$ |
| | | IDEAL | $81.1 \pm 1.9\%$ | $\mathbf{0.3} \pm 0.1\%$ | $91.4 \pm 0.3\%$ |
| 0.6% | JTT | WORST | $80.6 \pm 2.7\%$ | $\mathbf{8.7} \pm 1.4\%$ | $89.4 \pm 1.2\%$ |
| | | STANDARD | $79.3 \pm 1.7\%$ | $\mathbf{9.6} \pm 1.8\%$ | $89.0 \pm 0.8\%$ |
| | | IDEAL | $81.2 \pm 2.0\%$ | $\mathbf{0.5} \pm 0.1\%$ | $91.6 \pm 0.5\%$ |
| 0.8% | JTT | WORST | $78.5 \pm 3.4\%$ | $\mathbf{15.8} \pm 4.4\%$ | $87.8 \pm 1.2\%$ |
| | | STANDARD | $78.0 \pm 3.0\%$ | $\mathbf{16.8} \pm 5.4\%$ | $87.7 \pm 1.2\%$ |
| | | IDEAL | $81.2 \pm 1.5\%$ | $\mathbf{0.5} \pm 0.1\%$ | $91.0 \pm 0.6\%$ |
| 1% | JTT | WORST | $76.9 \pm 3.5\%$ | $\mathbf{20.9} \pm 7.9\%$ | $86.4 \pm 1.2\%$ |
| | | STANDARD | $78.0 \pm 4.1\%$ | $\mathbf{20.4} \pm 5.9\%$ | $86.7 \pm 1.2\%$ |
| | | IDEAL | $81.4 \pm 0.9\%$ | $\mathbf{0.5} \pm 0.3\%$ | $91.0 \pm 0.4\%$ |
| 2% | JTT | WORST | $64.7 \pm 7.4\%$ | $\mathbf{56.6} \pm 41.9\%$ | $69.8 \pm 5.6\%$ |
| | | STANDARD | $62.0 \pm 9.8\%$ | $\mathbf{59.8} \pm 45.9\%$ | $66.0 \pm 9.6\%$ |
| | | IDEAL | $82.5 \pm 1.6\%$ | $\mathbf{0.6} \pm 0.3\%$ | $91.1 \pm 0.2\%$ |
| 1% | GEORGE | WORST | $74.1 \pm 1.7\%$ | $\mathbf{16.5} \pm 2.9\%$ | $76.4 \pm 3.4\%$ |
| | | STANDARD | $77.3 \pm 2.5\%$ | $\mathbf{15.8} \pm 3.9\%$ | $79.4 \pm 1.6\%$ |
| | | IDEAL | $79.3 \pm 2.1\%$ | $\mathbf{0.4} \pm 0.0\%$ | $93.1 \pm 1.4\%$ |
| 2% | GEORGE | WORST | $74.3 \pm 5.4\%$ | $\mathbf{56.1} \pm 13.6\%$ | $78.0 \pm 9.0\%$ |
| | | STANDARD | $74.5 \pm 5.2\%$ | $\mathbf{52.5} \pm 7.1\%$ | $77.6 \pm 8.5\%$ |
| | | IDEAL | $80.2 \pm 0.4\%$ | $\mathbf{0.7} \pm 0.1\%$ | $92.4 \pm 0.9\%$ |

Table 9: Evaluating the impact of amplification in JTT on worst group accuracy (WGA), attack success rate (ASR) and test accuracy (ACC) when considering different targets for the GM attack.

| TARGETS | CASE | WGA | ASR | ACC |
|---|---|---|---|---|
| 5 | WORST | $76.1 \pm 3.2\%$ | $\mathbf{20.0} \pm 0.0\%$ | $89.9 \pm 1.0\%$ |
| 5 | STANDARD | $76.1 \pm 3.2\%$ | $\mathbf{20.0} \pm 0.0\%$ | $89.9 \pm 1.0\%$ |
| 5 | IDEAL | $75.9 \pm 0.8\%$ | $\mathbf{13.3} \pm 11.5\%$ | $91.3 \pm 0.1\%$ |
| 100 | WORST | $75.4 \pm 2.3\%$ | $\mathbf{8.6} \pm 0.5\%$ | $88.6 \pm 0.9\%$ |
| 100 | STANDARD | $75.8 \pm 2.2\%$ | $\mathbf{8.6} \pm 0.5\%$ | $88.8 \pm 1.0\%$ |
| 100 | IDEAL | $75.6 \pm 0.5\%$ | $\mathbf{5.6} \pm 0.5\%$ | $91.1 \pm 0.6\%$ |

Table 10: Evaluating the impact of amplification in JTT on worst group accuracy (WGA), attack success rate (ASR) and test accuracy (ACC) when considering a different architecture (ResNet-50). Note that ES denotes early stopping for the identification model.

| SETTING | ES | CASE | WGA | ASR | ACC |
|---|---|---|---|---|---|
| PRE-TRAINED | 60 | WORST | $72.4 \pm 2.1\%$ | $\mathbf{70.6} \pm 13.8\%$ | $76.2 \pm 1.3\%$ |
| | | STANDARD | $72.4 \pm 2.1\%$ | $\mathbf{70.6} \pm 13.8\%$ | $76.2 \pm 1.3\%$ |
| | | IDEAL | $85.1 \pm 1.4\%$ | $\mathbf{0.9} \pm 0.3\%$ | $90.2 \pm 1.0\%$ |
| PRE-TRAINED | 200 | WORST | $83.4 \pm 2.1\%$ | $\mathbf{58.0} \pm 23.7\%$ | $86.5 \pm 2.5\%$ |
| | | STANDARD | $83.5 \pm 1.9\%$ | $\mathbf{56.5} \pm 21.2\%$ | $86.7 \pm 2.1\%$ |
| | | IDEAL | $84.7 \pm 1.5\%$ | $\mathbf{0.5} \pm 0.5\%$ | $90.8 \pm 2.6\%$ |
| FROM SCRATCH | 200 | WORST | $39.5 \pm 7.8\%$ | $\mathbf{53.7} \pm 9.7\%$ | $59.0 \pm 9.8\%$ |
| | | STANDARD | $41.0 \pm 8.5\%$ | $\mathbf{34.5} \pm 26.4\%$ | $64.0 \pm 8.3\%$ |
| | | IDEAL | $37.8 \pm 6.9\%$ | $\mathbf{5.8} \pm 4.6\%$ | $61.0 \pm 12.9\%$ |

Table 11: Evaluating the impact of the poison base samples' choice for JTT and CelebA.

| Base Samples' Group | Gr. 0 (HRG) | Gr. 1 (HRG) | Gr. 2 (HRG) | Gr. 3 (LRG-1) |
|---|---|---|---|---|
| Label / Attribute | 0/0 | 0/1 | 1/0 | 1/1 |
| ASR | $97.4 \pm 0.1$ | $97.7 \pm 0.1$ | $91.2 \pm 2.3$ | $0.1 \pm 0.1$ |
| Loss Avg. (no poisons) | 0.16 | 0.07 | 0.79 | 1.90 |

Table 12: Evaluating the scenario when the attacker does not have any knowledge about the groups. We consider JTT and CelebA.

| | Base samples in class 0 | Base samples in class 1 |
|---|---|---|
| ASR (Worst) | $98.3 \pm 0.1$ | $97.6 \pm 0.3$ |
| ASR (Standard) | $90.7 \pm 1.7$ | $97.3 \pm 0.5$ |
| ASR (Ideal) | $0.1 \pm 0.0$ | $0.5 \pm 0.1$ |

Table 13: Evaluating the effect of JTT on CelebA with a different baseline (ERM) when varying the poison percentage. We consider the Subpopulation Attack (SA).

| Poison % | 0.05% | 0.1% | 0.2% | 0.3% | 0.5% | 1% | 2% |
|---|---|---|---|---|---|---|---|
| JTT (ASR) | 0.1% | 1.7% | 2.4% | 3.5% | 6.3% | 14.1% | 46.8% |
| ERM (ASR) | 0.1% | 0.1% | 0.2% | 0.0% | 0.1% | 0.1% | 0.3% |
| JTT (WGA) | 81.7% | 82.2% | 83.3% | 83.3% | 83.3% | 77.9% | 48.2% |
| ERM (WGA) | 43.3% | 43.3% | 47.2% | 42.8% | 41.1% | 45.0% | 45.6% |

Table 14: Evaluating the effect of AFR on CelebA with ERM as a baseline. We consider the Subpopulation Attack (SA).

| Poison % | 0.5% | 1% |
|---|---|---|
| AFR (ASR) | 2.1% | 14.4% |
| ERM (ASR) | 1.0% | 5.3% |
| AFR (WGA) | 75.3% | 78.4% |
| ERM (WGA) | 68.0% | 70.0% |

Table 15: The percentage of samples amplified by JTT from each group in Spawrious benchmark.

| Dataset | Gr.0 | Gr.1 | Gr.2 | Gr.3 | Gr.4 | Gr.5 | Gr.6 | Gr.7 | Poisons |
|---|---|---|---|---|---|---|---|---|---|
| M2M-easy | 4.9% | 4.1% | 2.5% | 3.1% | 4.8% | 6.6% | 2.1% | 3.5% | 99.6% |
| M2M-medium | 4.0% | 4.4% | 2.6% | 4.1% | 10.0% | 9.9% | 2.7% | 2.5% | 97.4% |
| M2M-hard | 4.1% | 2.7% | 4.0% | 4.3% | 3.4% | 4.3% | 4.6% | 3.9% | 100% |

Table 16: The impact of poisoning defenses in federated learning on group robustness.

| METHOD | IID? | WGA DROP | ACC DROP |
|---|---|---|---|
| MEDIAN | YES | $23.7 \pm 7.3\%$ | $-0.2 \pm 0.3\%$ |
| TRIMMED MEAN | YES | $45.7 \pm 8.8\%$ | $10.6 \pm 3.5\%$ |
| SPARSEFED ($\rho = 0$) | YES | $45.4 \pm 27.7\%$ | $11.0 \pm 17.7\%$ |
| SPARSEFED ($\rho = 0.9$) | YES | $42.7 \pm 24.3\%$ | $19.0 \pm 14.6\%$ |
| SPARSEFED ($\rho = 0.9$) | No | $55.1 \pm 21.6\%$ | $12.3 \pm 16.3\%$ |

Table 17: The impact of EPIc on group robustness, when considering the DLBD attack and several poison percentages.

| POISONS | CASE | WGA | ASR | ACC |
|---|---|---|---|---|
| | IDEAL | $\mathbf{61.2} \pm 2.1\%$ | $0.1 \pm 0.0\%$ | $95.4 \pm 0.5\%$ |
| 0.5% | STANDARD | $\mathbf{57.6} \pm 4.9\%$ | $0.1 \pm 0.0\%$ | $95.0 \pm 0.6\%$ |
| | WORST | $\mathbf{52.0} \pm 8.1\%$ | $0.1 \pm 0.0\%$ | $94.4 \pm 0.5\%$ |
| | IDEAL | $\mathbf{59.0} \pm 2.5\%$ | $0.1 \pm 0.1\%$ | $95.5 \pm 0.1\%$ |
| 1% | STANDARD | $\mathbf{55.8} \pm 6.0\%$ | $0.1 \pm 0.0\%$ | $95.0 \pm 0.1\%$ |
| | WORST | $\mathbf{50.7} \pm 6.9\%$ | $0.1 \pm 0.0\%$ | $94.3 \pm 0.8\%$ |
| | IDEAL | $\mathbf{57.3} \pm 2.5\%$ | $0.3 \pm 0.3\%$ | $94.5 \pm 0.6\%$ |
| 2% | STANDARD | $\mathbf{57.0} \pm 0.4\%$ | $0.5 \pm 0.4\%$ | $94.5 \pm 0.4\%$ |
| | WORST | $\mathbf{48.6} \pm 7.6\%$ | $0.2 \pm 0.0\%$ | $93.8 \pm 0.1\%$ |

robust aggregation mechanisms are widely studied. In this scenario, the defense does not sanitize the training set but sanitizes the updates sent by each client to prevent poisoning. We consider FedAvg (McMahan et al., 2017) as an un-defended baseline and study the drop in WGA and ACC relative to it. We included more details about the experimental setup in Appendix A.1. In Table 16, we observe that the defenses we consider cause significantly more drop in WGA than in ACC, over the baseline. This exposes that all these methods, while attempting to fight against poisoning, end up having a disparate impact on the model's accuracy on under-represented groups.

**More Settings.** In Table 17, we study the impact of EPIc when there are $0.5\%$ or $2\%$ poisons for the DLBD attack, instead of $1\%$. We observe that EPIc drops the WGA by $0.3\% - 3.6\%$, while maintaining a low ASR and high ACC. Also, aligning with our previous results, we observe that the overall WGA is low compared to the values obtained when considering a group robustness method. In Table 18, we run additional experiments for EPIc on CelebA with $20\%$, $50\%$ and $100\%$ of the dataset considered. The persistent gap between the Ideal and Standard cases demonstrates the negative impact of EPIc on group robustness. Overall, these results are consistent with our main claims.

**Run-time defenses.** Additionally, we run experiments using STRIP (Gao et al., 2019), a run-time backdoor detection mechanism. The main assumption of this method is that a backdoored model's outputs have lower entropy on perturbed triggered samples compared to perturbed clean samples. In Figure 3, we show the percentage of samples from each group which are detected by the defense when varying the entropy threshold (we use the first model from Table 2 in the standard case, run the experiment 3 times and provide the mean of the results). Basically, the result from Figure 3 shows that there is no entropy threshold for which STRIP can detect the triggered samples without inadvertently detecting the clean samples, too. We believe that the strong regularization needed for the models to achieve group robustness (Liu et al., 2021; Sagawa et al., 2019) contributes to the limitation of such defenses as the model does not have very confident outputs, leading to generally high entropy values for all types of samples.

## A.5 COMBINING GROUP ROBUSTNESS METHODS AND POISONING DEFENSES

In this section, we study the feasibility of achieving both high group robustness and poisoning resilience by combining the current state-of-the-art methods.

Table 18: The impact (in terms of WGA) of EPIc on group robustness, when considering the DLBD attack and several subset sizes of CelebA.

|  | 20% OF CELEBA | 50% OF CELEBA | FULL CELEBA |
|---|---|---|---|
| IDEAL | $52.3 \pm 1.6$ | $50.3 \pm 1.9$ | $45.9 \pm 5.8$ |
| STANDARD | $43.1 \pm 5.5$ | $46.4 \pm 2.8$ | $43.6 \pm 3.0$ |
| WORST | $32.3 \pm 2.7$ | $32.0 \pm 0.8$ | $37.7 \pm 3.8$ |

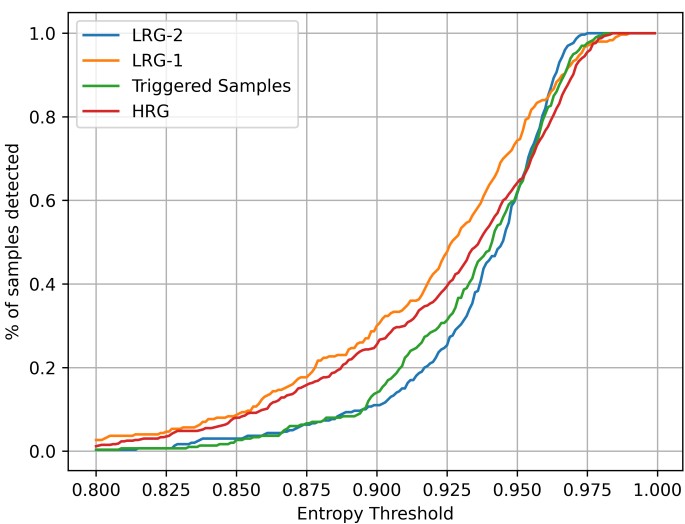

Figure 3: The percentage of samples from each group detected by STRIP when varying the threshold.

We first apply EPIc to identify potential poisons in the training set. Then, when we apply JTT, we intervene so that it does not amplify the potential poisons found in its first phase (i.e., we remove them from JTT's upsampling set). We have considered the following two baselines in our pipeline: an ideal EPIc that identifies only the poisons and not using EPIc that we would want to improve upon. Because EPIc removes samples iteratively, we considered three stopping epochs for the removal process, so that we have control over how many samples EPIc identifies as poisons. As shown in Figure 2, stopping EPIc sooner leads to a lower percentage of samples from LRG that are removed, but also a lower amount of poisons. Whereas, stopping EPIc later increases both of these rates.

For the rest of this section, we consider Waterbirds dataset and DLBD attack. More details on the experimental setup, including the hyper-parameters used are in Appendix A.1.

First of all, as shown in Table 19, not using EPIc at all results in a WGA relatively close to the WGA that could be obtained if none of the poisons were amplified (Ideal EPIc). However, the ASR is still high if we do not use EPIc to identify potential poisons. We evaluate three possible stopping epochs for EPIc and measure the effects on WGA and ASR as a function of EPIC's stopping epoch. With a higher stopping epoch (i.e., more samples are identified as poisons), the ASR decreases, however, the WGA also decreases. For example, to mitigate the attack and obtain below 1% ASR, we need to sacrifice over 35% WGA—significant damage to group robustness. Also, in ideal EPIc (only poisons are eliminated), we could obtain both high WGA (over 80%) and low ASR (lower than 1%). Moreover, note that for all the models, the ACC stays relatively high (over 80%), though, lower than the settings without any poisoning (e.g., 93.3% in Liu et al. (2021)). This further shows how ACC can be misleading to judge the side effects of a poisoning defense.

In conclusion, we have attempted to combine EPIc and JTT, in hopes of achieving both high poisoning resilience and group robustness, but this task is not trivial. Both legitimate under-represented

Table 19: Applying EPIc and JTT together to combine poison resilience with group robustness.

| EPIC | WGA | ASR | ACC |
|------|-----|-----|-----|
| No | $78.0 \pm 4.1\%$ | $20.4 \pm 5.9\%$ | $86.7 \pm 1.2\%$ |
| stop = 3 | $76.5 \pm 3.8\%$ | $14.8 \pm 4.0\%$ | $91.9 \pm 0.3\%$ |
| stop = 5 | $73.4 \pm 7.4\%$ | $6.7 \pm 1.7\%$ | $82.8 \pm 7.3\%$ |
| stop = 7 | $42.3 \pm 6.6\%$ | $0.9 \pm 0.6\%$ | $93.4 \pm 0.2\%$ |
| Ideal | $81.4 \pm 0.9\%$ | $0.5 \pm 0.3\%$ | $91.0 \pm 0.4\%$ |

samples and poison samples in realistic attacks can be difficult-to-learn and without making specific assumptions, (e.g., poisons contain detectable artifacts), it might be difficult to distinguish them. Using EPIc (which makes no such assumptions) to identify potential poisons and use that information as an intervention into JTT is not enough to mitigate the trade-off between WGA and ASR.

### A.6 Proofs

**Lemma 4.1** (Restated) *For the setting described above, if we assume that there are no ties in maximum expected class probability among groups, then the identification model has less expected class probability on the poisons* $(y_m, a_m)^*$ *in comparison to any legitimate group.*

*Proof.* For any $y, a \in \{0, 1\}$, we know $p_{y_m, a_m} > p_{y,a}^*$, so

$$1 - p_{y_m, a_m} < 1 - p_{y,a}^*. \tag{1}$$

Also, for any $y, a \in \{0, 1\}$, from the definition of $p^*$, we have:

$$p_{y,a} + p_{y,a}^* = 1 \tag{2}$$

$(\mathbb{E}_{(x,y,a) \sim (D_{(y,a)}, y, a)}[I(x)_y] + \mathbb{E}_{(x,1-y,a) \sim (D_{(y,a)}, 1-y, a)}[I(x)_{1-y}] = \mathbb{E}_{(x,y,a) \sim (D_{(y,a)}, y, a)}[I(x)_y + I(x)_{1-y}] = \mathbb{E}_{x \sim (D_{(y,a)}, y, a)}[I(x)_0 + I(x)_1] = \mathbb{E}_{x \sim (D_{(y,a)}, y, a)}[1] = 1).$

Hence, by substituting 2 in 1, we obtain:

$$p_{y_m, a_m}^* < p_{y,a}. \tag{3}$$

Therefore, the identification model has the least expected class probability on the poison samples $(y_m, a_m)^*$. $\square$

**Theorem 4.2** (Restated) *We consider the same setting as in Lemma 4.1. We denote the poisons* $(y_m, a_m)^*$ *by* $g_p$ *and let* $(y, a) := g_c$ *be any group of samples (e.g., a legitimate minority group). Also, we denote* $I(x)_{1-y_m}$ *for* $(x, 1 - y_m, a_m) \sim (D_{g_p}, 1 - y_m, a_m)$ *by* $G_p$ *and* $I(x)_y$ *for* $(x, y, a) \sim (D_{g_c}, y, a)$ *by* $G_c$ *and the cross-entropy loss on* $G \in \{G_c, G_p\}$ *by* $L(G)$. *We assume* $G_p$ *and* $G_c$ *are independent and* $Var(G_p) = Var(G_c) := \sigma^2$ *(i.e., the variances of the class probability for the identification model are equal for the legitimate group and for the poisons) and denote* $\mathbb{E}(G_p) := \mu_p$ *and* $\mathbb{E}(G_c) := \mu_c$. *Then, for any* $\epsilon \in (0, 1)$, *if* $\sigma \leqslant \sqrt{\frac{1}{\sqrt{1-\epsilon}} - 1} \cdot \frac{\mu_c - \mu_p}{2}$, *we have* $\mathbb{P}(L(G_c) < L(G_p)) > 1 - \epsilon$.

*Proof.* Let $\epsilon \in (0, 1)$. We observe that $\mathbb{P}(L(G_c) < L(G_p)) = \mathbb{P}(-log(G_c) < -log(G_p)) = \mathbb{P}(G_p < G_c) > \mathbb{P}(G_p < \frac{\mu_c + \mu_p}{2} < G_c) = \mathbb{P}(G_p < \frac{\mu_c + \mu_p}{2}) \wedge G_c > \frac{\mu_c + \mu_p}{2})) = \mathbb{P}(G_p < \frac{\mu_c + \mu_p}{2}) \cdot \mathbb{P}(G_c > \frac{\mu_c + \mu_p}{2}) = \mathbb{P}(G_p - \mu_p < \frac{\mu_c - \mu_p}{2}) \cdot \mathbb{P}(G_c - \mu_c > -\frac{\mu_c - \mu_p}{2}) = (1 - \mathbb{P}(G_p - \mu_p \geqslant \frac{\mu_c - \mu_p}{2})) \cdot (1 - \mathbb{P}(G_c - \mu_c \leqslant -\frac{\mu_c - \mu_p}{2})).$

We know from Lemma 4.1 that $\mu_p < \mu_c$, (i.e., $\frac{\mu_c - \mu_p}{2} > 0$), so we can apply Cantelli's inequality to obtain: $(1 - \mathbb{P}(G_p - \mu_p \geqslant \frac{\mu_c - \mu_p}{2})) \cdot (1 - \mathbb{P}(G_c - \mu_c \leqslant -\frac{\mu_c - \mu_p}{2})) \geqslant (1 - \frac{\sigma^2}{\sigma^2 + (\frac{\mu_c - \mu_p}{2})^2}) \cdot (1 - \frac{\sigma^2}{\sigma^2 + (\frac{\mu_c - \mu_p}{2})^2}) = \frac{(\frac{\mu_c - \mu_p}{2})^2}{\sigma^2 + (\frac{\mu_c - \mu_p}{2})^2} \cdot \frac{(\frac{\mu_c - \mu_p}{2})^2}{\sigma^2 + (\frac{\mu_c - \mu_p}{2})^2}$, so $\mathbb{P}(L(G_c) < L(G_p)) > \frac{(\frac{\mu_c - \mu_p}{2})^2}{\sigma^2 + (\frac{\mu_c - \mu_p}{2})^2} \cdot \frac{(\frac{\mu_c - \mu_p}{2})^2}{\sigma^2 + (\frac{\mu_c - \mu_p}{2})^2}.$

Since we know $\sigma \leqslant \sqrt{\frac{1}{\sqrt{1-\epsilon}} - 1} \cdot \frac{\mu_c - \mu_p}{2}$, we can conclude that $\mathbb{P}\big(L(G_c) < L(G_p)\big) >$

$$\frac{(\frac{\mu_c - \mu_p}{2})^2}{\left(\frac{1}{\sqrt{1-\epsilon}} - 1\right) \cdot (\frac{\mu_c - \mu_p}{2})^2 + (\frac{\mu_c - \mu_p}{2})^2} \cdot \frac{(\frac{\mu_c - \mu_p}{2})^2}{\left(\frac{1}{\sqrt{1-\epsilon}} - 1\right) \cdot (\frac{\mu_c - \mu_p}{2})^2 + (\frac{\mu_c - \mu_p}{2})^2} = \left(\frac{1}{\frac{1}{\sqrt{1-\epsilon}} - 1 + 1}\right)^2 = 1 - \epsilon. \qquad \square$$

