# OpenReview forum: "Like Oil and Water: Group Robustness Methods and Poisoning Defenses May Be at Odds"
_ICLR.cc/2024/Conference — ICLR 2024 poster_

### Official Review · Reviewer_Fmsc · 2023-10-27

**Soundness:** 2 fair
**Presentation:** 2 fair
**Contribution:** 4 excellent
**Rating:** 8
**Confidence:** 3

**Summary:**

This paper studies the tension between group robustness methods and poisoning defenses; in particular, the authors show that group robustness methods which pseudo-label the minority group are often unable to distinguish minority samples from poison samples. On the other hand, poisoning defenses can make the minority group more difficult to learn, which hurts group robustness. The conclusions are supported by a variety of experiments and a simple theoretical result.

**Strengths:**

1. This paper makes an important contribution by exposing the incompatibility of current group robustness and poisoning defenses, especially showing that group robustness methods can make models more susceptible to poisoning adversaries. This observation is novel as far as I know, and should be interesting for the community.
2. The experiments use a variety of robustness and poisoning techniques and metrics (if not datasets) and the extension to a federated learning application is welcome. The authors also provide a theoretical result in a simple yet well-motivated setting.
3. The authors show that a naive combination of both techniques fails to resolve their tension, an interesting result which could represent a new direction of research for the community.

**Weaknesses:**

The main weakness of the paper is its limited dataset evaluation, which consists only of Waterbirds and a 10% subset of CelebA. While the experiments are indeed comprehensive with different techniques and metrics, the size and composition of the datasets questions whether the results hold more generally.

In particular, Waterbirds is known to the community to be a limited benchmark and should mostly be used for sanity checking or simple comparisons. First, Waterbirds is an easy benchmark in the sense that simple class balancing methodologies using ERM achieve similar worst-group accuracy performance to methods like Group DRO which utilize group information (as close as 1% WGA in [1]). Second, the validation and test distributions of Waterbirds are more amicable compared to the training dataset, as they are group-balanced conditioned on the classes; this can skew model selection in some cases. Third, Waterbirds is known to contain incorrect labels as well as images with both landbirds and waterbirds present [2]. Finally, while CelebA is a more complex dataset than Waterbirds, and the 10% subset was likely necessary due to computational restrictions, the fact remains that both datasets are small, belong to the vision domain, and only utilize binary-valued spurious features.

Some suggestions may include Spawrious [3] for the vision domain (in particular because it is not too large; it is a difficult dataset which is smaller than CelebA) or MultiNLI [4] for the language domain (which would also introduce a non-binary-valued spurious feature).

I also found the writing unclear in some parts, detailed in the next section.

**Questions:**

1. I would encourage the authors to perform a more detailed literature review, as the related work subsection for group robustness is out of date and does not include any references more recent than 2021. There have been significant contributions since 2021, particularly in methods which pseudo-label the minority group (with or without an explicit identification model), e.g., [2,5,6,7,8,9].
2. The definitions in Section 2.2 are unclear and should be improved.
    * I believe the reason that the definitions of groups differs from [4] is to account for the dirty-label samples later in Section 3.3. This should be explicitly stated prior to the definitions, as otherwise the definitions seem inconsistent with the literature. Is the worst-group accuracy in the remainder of the paper computed with respect to these new groups (i.e., 8 groups) or the original groups (i.e., 4 groups)? This should also be made explicit.
    * The definition of a minority group seems somewhat arbitrary and inconsistent with the literature. As far as I understand the definition given is that for a dataset of size $n$, a group $g\in G_Y$ is a minority group iff the number of data belonging to $g$ is less than $n/|G_Y|$. If we use 4 groups for CelebA as is common in the literature, this would mean that blond-female is a minority group in CelebA (it has 22880 datapoints while the total data is 162770 over 4 groups), whereas the text states that the only minority group is blond-male. On the other hand, if we use 8 groups as I believe this work does, then blond-female is barely a majority group, and hence it is not clear that it remains a majority group when a random 10% subset of CelebA is taken.
3. The bibtex could use an update: there are some extra braces and capitalization errors.
4. It would be helpful for reference to include a table with the proportions of each group and class in each benchmark dataset, as well as the proportions of each group and class in the 10% sample of CelebA used in the experiments. See [9, 10] for examples.
5. The definitions in Section 3.3 are unclear and should be improved. First, it should be made clear that $I(x)\in [0,1]$, i.e., that it is a probability output rather than a logit output. Second, $I(x)_y$ is referred to as the “confidence” of model $I$ on input $x$ for class $y$, which is misleading: if $I(x)_y=0$ the model is in fact very confident (that $x$ does not belong to class $y$). I would expect the definition of “confidence” in this case to look more like $2|I(x)_y - 1/2|$, since $1/2$ can be considered as the “least confident” prediction in a binary classification problem.
6. In Appendix A.2, should “DLDB” read “DLBD”?
7. In Figure 1, it should be made clear that the lighter-colored circles and triangles are amplified points (it is otherwise an excellent figure). Also, the two shades of red are indistinguishable.
8. The phrase “this boost is almost as high as it could have been” in Section 3.2 is confusing and could use a rephrase, perhaps connecting it to the worst-case approach tested earlier.
9. In Section 5, “conciliate” should perhaps be replaced by “reconcile”.
10. At the bottom of page 7, should the citation to Wu et al. be in parentheses, i.e., \citep?

***Recommendation***

Overall, the novelty and importance of the contribution cause me to lean slightly more towards acceptance than rejection, but I believe the paper would be greatly improved with additional results on more rigorous benchmark datasets as detailed in the Weaknesses section, as well as improvements to the clarity of the writing.

***After author discussion***

In light of the authors' response, which addressed all my concerns, I have concluded that this paper is valuable for the community and should be accepted to ICLR. Therefore, I have raised my score from a 6 to an 8.

***References***

[1] Idrissi et al. Simple data balancing achieves competitive worst-group-accuracy. CLeaR, 2022.

[2] Taghanaki et al. MaskTune: Mitigating Spurious Correlations by Forcing to Explore. NeurIPS, 2022.

[3] Lynch et al. Spawrious: A Benchmark for Fine Control of Spurious Correlation Biases. ArXiv, 2023.

[4] Sagawa et al. Distributionally Robust Neural Networks for Group Shifts: On the Importance of Regularization for Worst-Case Generalization. ICLR, 2020.

[5] Kim et al. Learning Debiased Classifier with Biased Committee. NeurIPS, 2022.

[6] Sohoni et al. BARACK: Partially Supervised Group Robustness With Guarantees. ICML SCIS Workshop, 2022.

[7] Zhang et al. Correct-N-Contrast: A Contrastive Approach for Improving Robustness to Spurious Correlations. ICML, 2022.

[8] Qiu et al. Simple and Fast Group Robustness by Automatic Feature Reweighting. ICML, 2023.

[9] LaBonte et al. Towards Last-layer Retraining for Group Robustness with Fewer Annotations. NeurIPS, 2023.

[10] Kirichenko et al. Last Layer Re-Training is Sufficient for Robustness to Spurious Correlations. ICLR, 2023.

---

> ### Author Response · Authors · 2023-11-23
> **Response to Reviewer Fmsc (1)**
>
> We thank the reviewer for their very constructive and detailed feedback! Please find below our response to your comments:
>
> > The main weakness of the paper is its limited dataset evaluation, which consists only of Waterbirds and a 10% subset of CelebA. While the experiments are indeed comprehensive with different techniques and metrics, the size and composition of the datasets questions whether the results hold more generally. [...] Finally, while CelebA is a more complex dataset than Waterbirds, and the 10% subset was likely necessary due to computational restrictions, the fact remains that both datasets are small, belong to the vision domain, and only utilize binary-valued spurious features.
>
> To address the reviewer's concern, we ran the experiments from Table 3 on different sizes of CelebA (all the other experiments in the paper were already on full CelebA). Below, we show the effect of EPIc (poisoning defense) on the WGA (results are over 3 runs):
>
> |    |20% of CelebA  | 50% of CelebA | Full CelebA      |
> |----|---------------|---------------|---------------|
> |   Ideal  | 52.3 +- 1.6  | 50.3 +- 1.9   | 45.9 +- 5.8 |
> | Standard  | 43.1 +- 5.5  | 46.4 +- 2.8   | 43.6 +- 3.0|
> |   Worst  | 32.3 +- 2.7  | 32.0 +- 0.8   |  37.7 +- 3.8|
>
> This shows that the trends we observed (i.e., EPIc hurts the WGA by eliminating minority groups) are consistent as the ideal WGA (when we intervene to prevent EPIc from eliminating any minority samples) is higher than the standard WGA (applying EPIc w/o any intervention).
>
> Additionally, to show the amplification due to group robustness methods, we ran experiments with the Subpopulation Attack (SA) on full CelebA for JTT and ERM (the baseline suggested by Reviewer Shsu).
>
> |Poison %      |      0.05%   |       0.1%   |        0.2%   |   0.3%  |    0.5% |      1% |          2%|
> |--------------------|-----------|-----------|----------|-----------|-----------|-----------|-----------|
> |JTT (ASR)   |         0.1%  |      1.7%  |          2.4% |     3.5%    |    6.3%   |   14.1%  |    46.8%|
> |ERM (ASR)  |  0.1%   |     0.1%         |  0.2%   |    0.0%   |    0.1%     |  0.1%       |  0.3%|
>
>
> | Poison %       |  0.05%   |        0.1%     |      0.2%   |      0.3%  |      0.5%  |     1%   |       2%|
> | ------------------- | -----------  | ----------- |----------|-----------|-----------|-----------|-----------|
> |JTT (WGA)    |     81.7%  |    82.2%   |      83.3%  |      83.3% |    83.3%  |  77.9% |  48.2%|
> |ERM (WGA) |     43.3%    | 43.3%   |       47.2%   |    42.8%  |   41.1%  |  45.0% |  45.6%|
>
> In the first table, we observe that the amplification is generally high (a large ASR gap between JTT and ERM). The second table shows that JTT still works as intended by improving the WGA over ERM. In conclusion, our observations remain consistent for the full CelebA as well.
>
> > Some suggestions may include [...] MultiNLI [4] for the language domain (which would also introduce a non-binary-valued spurious feature). [...] There have been significant contributions since 2021, particularly in methods which pseudo-label the minority group (with or without an explicit identification model), e.g., [2,5,6,7,8,9]
>
> Thank you for suggesting these experiments! To ensure the validity of our observations for newer group robustness methods and more advanced datasets, we applied AFR [1] (referenced as [8] by the reviewer) on MultiNLI. As in our previous experiment, we consider SA as the attack, ERM as a baseline (no group robustness method), and report both the ASR and WGA.
>
> | Poison %          |    0.5%     |      1%     |
> | ------------------- | -----------  | ----------- |
> |AFR (ASR)       |    2.1%       |  14.4%      |
> |ERM (ASR) |   1.0%      |     5.3%    |
>
>
> | Poison %          |    0.5%      |     1%     |
> | ------------------- | -----------  | ----------- |
> |AFR (WGA)        |   75.3%     |      78.4%  |
> |ERM (WGA)  |  68.0%      |     70.0%   |
>
> We observe the same tension in these experiments: while AFR improves the WGA over the baseline (as expected), it also inadvertently amplifies the ASR (e.g., from 5.3% to 14.4%, a significant boost for a weak attacker).
>
> > Some suggestions may include Spawrious [3] for the vision domain [...].
>
> We agree with the reviewer that Spawrious would be a good addition to our experiments. Unfortunately, due to time limitations, we were unable to conclude the experiments on this benchmark before the rebuttal deadline. However, we are committed to include them in the final version (we will include them in the Appendix).
>
> > I would encourage the authors to perform a more detailed literature review [...]
>
> We will update our paper's appendix with the discussion we shared with Reviewer Shsu on the different (more recent) group robustness methods (some of them are referenced as [5,7,8,9] by the reviewer), and the parallels among the heuristics they rely on.

---

> ### Author Response · Authors · 2023-11-23
> **Response to Reviewer Fmsc (2)**
>
> > The definitions in Section 2.2 are unclear and should be improved [...] The definitions in Section 3.3 are unclear and should be improved.
>
> We agree with the reviewer. We included more clear definitions for groups, as well for the identification model I(x) in the paper.
>
> > [Comments about the bibtex, styling errors, colors in the figures, confusing phrases, and other typos]
>
> Thanks for pointing these out, we fixed all of them!
>
> > It would be helpful for reference to include a table with the proportions of each group and class in each benchmark dataset, as well as the proportions of each group and class in the 10% sample of CelebA used in the experiments.
>
> We agree, we will include it in an updated version of the paper.
>
>
> [1] Simple and Fast Group Robustness by Automatic Feature Reweighting, Qiu et al. (ICML'23)

---

> > ### Comment · Reviewer_Fmsc · 2023-11-30
> > **Thank you for the response**
> >
> > Thanks to the authors for their very thorough response; all of my concerns have been addressed. In light of this, and after reading the other reviews and responses, I have concluded that this work is valuable for the community and should be accepted to ICLR. Therefore, I have raised my score from a 6 to an 8.
> >
> > Some short final comments:
> > 1. The preliminary experiments on MultiNLI and AFR are very interesting and I look forward to reading the detailed results in the revised version.
> > 2. "all the other experiments in the paper were already on full CelebA": I would encourage the authors to clarify this in the paper as it was not clear to me on first reading.
> > 3. I agree with Reviewer a2i2 that a title change may be prudent to set proper expectations for the reader. Minor nitpick: I would suggest "may be at odds" instead of "are at odds" in the authors' suggested title. I feel it's more appropriate for this paper for the title to be observatory rather than conclusive, but perhaps this is up to personal preference.

---

### Official Review · Reviewer_Shsu · 2023-10-30

**Soundness:** 2 fair
**Presentation:** 2 fair
**Contribution:** 1 poor
**Rating:** 5
**Confidence:** 4

**Summary:**

The authors consider the interplay between a certain type of group robustness methods and poisoning attacks. They show that methods to achieve high accuracy on minority groups in the absence of (validation) group annotations also often flag poisoned examples, and thus amplify poisoning attacks. Additionally, they show that poisoning defenses often flag minority examples as poisons, thus removing them from the dataset. Finally, the authors complement their experimental findings with an formal impossibility result in a toy setting.

**Strengths:**

The authors present an interesting connection/trade-off between two fields within ML. The problem of exploring trade-offs between competing objective is important.

**Weaknesses:**

### Overclaiming / limited scope of conclusions
I believe that the statement
"We show experimentally that group robustness methods fail to distinguish minority groups from poisons."
is misleading. In particular, the authors only consider group robustness methods which use a loss-based heuristic in lieu of *validation* group annotations. This significantly reduces the scope of authors' contributions (to the best of my knowledge, at the time of writing, only ~3 group robustness methods use the above heuristic), and in my opinion is not properly reflected in the writing.

### No evidence the phenomenon persists for attacks with high efficacy (success rate)

My main concern with the paper is the following: if I have an attack (e.g., backdoor attack) with very high ASR, I would expect to have low loss on poisoned (e.g., backdoored) examples in the validation set. Hence, there's no reason for me to expect that JTT/GEORGE/etc would flag them as minority examples. In most of the reported setups, e.g. Table 2, the attack success rate is very low (e.g., 20%), even in the "worst" case when the group robustness method amplifies the attack.


Minor point: It would have been nice to report results for the state of the art method (to the best of my knowledge) that uses the loss-based heuristic [1].

[1] Qiu, Shikai, Andres Potapczynski, Pavel Izmailov, and Andrew Gordon Wilson. "Simple and Fast Group Robustness by Automatic Feature Reweighting." arXiv preprint arXiv:2306.11074 (2023).

**Questions:**

### Is ASR actually amplified by group robustness methods

I am confused the statement "The large gap between (6.7% ´ 97.4%) the standard and ideal cases shows an opportunity for better heuristics" (referring to Table 2). It is "not the job" of the group robustness method to remove the poisoning attack. Thus, a more fair comparison in my opinion would be to compare "standard" ASR against the baseline of not applying the group robustness method. What is the gap then?

---

> ### Author Response · Authors · 2023-11-23
> **Response to Reviewer Shsu (1)**
>
> We thank the reviewer for their constructive feedback! Please find below our response to each of your comments:
>
> > …the authors only consider group robustness methods which use a loss-based heuristic in lieu of validation group annotations. This significantly reduces the scope of authors' contributions…
>
> Although we experimented with methods that specifically use a loss-based heuristic, we believe that many other group robustness methods rely on similar heuristics, perhaps less directly. Here, we discuss four methods from recent literature that implemented seemingly different heuristics that will align with loss-based heuristics:
>
> * **Correct-N-Contrast: A Contrastive Approach for Improving Robustness to Spurious Correlations, Zhang et al. (ICLR’22)**: This method applies contrastive learning to push together the representations of training samples that are labeled into the same class but predicted differently. Since, the low-budget poisoning attacks we consider often generate hard-to-learn, misclassified samples, we believe this method will also suffer from the tension we identify, e.g., misclassified poison samples of class $y$  will be represented similarly to correctly classified samples of class $y$, which will amplify their effectiveness as the model becomes less likely to misclassify them.
>
> * **Dropout Disagreement: A Recipe for Group Robustness with Fewer Annotations, LaBonte et al. (NeurIPS’22 Workshop)**: This method uses the disagreements between different forward passes of the model with dropout to find which training samples to amplify. Different dropout forward passes are known to disagree on samples where the model has high uncertainty [1], which are often hard-to-learn, higher-loss samples. As a result, we believe this method will also suffer from the same tension we identified, e.g., the hard-to-learn poison samples we generated will create dropout disagreements and will be amplified.
>
> * **Learning Debiased Classifier with Biased Committee, Kim et al. (NeurIPS’22)**: This method relies on disagreements among the members of an ensemble of models to identify the “biased” samples (i.e., the samples that contain spurious correlations). If a sample has low ensemble agreement, it will be amplified by this method. Ensemble disagreement is a known uncertainty metric [2] in the literature. As a result, we believe that this method will also suffer from the same tension we identified, e.g., the hard-to-learn poison samples will be misclassified by more members of the ensemble, and will be amplified.
>
> * **Towards Last-layer Retraining for Group Robustness with Fewer Annotations, LaBonte et al. (NeurIPS’23)**: This method constructs a reweighting set based on either (1) ERM model’s misclassifications, or (2) the misclassifications of an early-stopping model, or (3) dropout disagreement (similar to the Dropout Disagreement paper) or (4) the disagreements between the ERM and early-stopping models. All these heuristics to identify minority group samples will end up identifying the hard-to-learn poisoning samples as well. The authors identified (4) as the most promising heuristic. It is known that hard-to-classify samples are learned at later iterations of model training [3]. As a result, we believe that method will also suffer from the same tension we identified, e.g., the hard-to-learn poison samples will be classified differently by the early-stopping and regular models, and will be amplified.
>
>
> These common threads show that, despite not using the loss heuristic directly, many methods in this line of work rely on related ideas that will amplify uncertain, misclassified, high-loss, or hard-to-learn samples. Moreover, it is known that many example difficulty metrics are highly correlated with one another [3,4]. Ultimately, we believe that these related heuristics cannot avoid the tension we identified because low-budget poisoning attacks generate hard-to-learn samples. To address the reviewer's concern and communicate that our findings have wider implications than JTT/GEORGE, we will add this discussion to our paper (we plan to add it in the Appendix). Also, please find below our new experiments on AFR [7], a state-of-the-art group robustness method.

---

> > ### Author Response · Authors · 2023-11-23
> > **Response to Reviewer Shsu (2)**
> >
> > > …if I have an attack (e.g., backdoor attack) with very high ASR, I would expect to have low loss on poisoned (e.g., backdoored) examples in the validation set. Hence, there's no reason for me to expect that JTT/GEORGE/etc would flag them as minority examples. In most of the reported setups, e.g. Table 2, the attack success rate is very low (e.g., 20%), even in the "worst" case when the group robustness method amplifies the attack…
> >
> > We fully agree with the reviewer's assessment. An already powerful poisoning attack with a high ASR will not be flagged or amplified. However, we are focusing on adversaries who cannot achieve high ASR because they have a limited budget (i.e., the number of injected samples).  A realistic attacker aims to achieve the highest ASR possible within their limited budget (e.g., each poison requires the adversary to compromise a website to inject an image [5]).
> >
> > The main takeaway from our experiments is that the defender’s use of group robustness methods might provide a much-needed boost to a low-budget attacker (who cannot achieve a high ASR because of this low budget). This type of amplification is known to be attractive for real-world attackers, for example in Denial-of-Service attacks [6], as they get “more bang for their buck” due to the practices of the defender.
> >
> > > Question: ..Thus, a more fair comparison in my opinion would be to compare "standard" ASR against the baseline of not applying the group robustness method. What is the gap then?
> >
> > To address the reviewer's concerns, we considered the Subpopulation Attack (SA) on CelebA when applying JTT and without applying any group robustness method (ERM) as a baseline. The two tables below show the ASR and WGA, respectively.
> >
> > |Poison %      |      0.05%   |       0.1%   |        0.2%   |   0.3%  |    0.5% |      1% |          2%|
> > |--------------------|-----------|-----------|----------|-----------|-----------|-----------|-----------|
> > |JTT (ASR)   |         0.1%  |      1.7%  |          2.4% |     3.5%    |    6.3%   |   14.1%  |    46.8%|
> > |ERM (ASR)  |  0.1%   |     0.1%         |  0.2%   |    0.0%   |    0.1%     |  0.1%       |  0.3%|
> >
> >
> > | Poison %       |  0.05%   |        0.1%     |      0.2%   |      0.3%  |      0.5%  |     1%   |       2%|
> > | ------------------- | -----------  | ----------- |----------|-----------|-----------|-----------|-----------|
> > |JTT (WGA)    |     81.7%  |    82.2%   |      83.3%  |      83.3% |    83.3%  |  77.9% |  48.2%|
> > |ERM (WGA) |     43.3%    | 43.3%   |       47.2%   |    42.8%  |   41.1%  |  45.0% |  45.6%|
> >
> > In the first table, we observe that the amplification is generally high (large gap between JTT and the baseline). The second table shows that JTT still works as intended as it improves the WGA over the baseline.
> >
> >
> > Furthermore, we would like to clarify that an ASR of 14% (e.g., for 1% poisons) for the SA indicates a **relative accuracy drop** on the attack's target group over a non-poisoned model. This is a significant drop, especially compared to the 0.1% drop against the baseline method (ERM) that doesn't employ any group robustness technique.
> >
> >
> > > …report results for the state of the art method (to the best of my knowledge) that uses the loss-based heuristic…
> >
> > To address the reviewer’s concerns, we consider AFR [7] (referenced by the reviewer as [1]) on MultiNLI (the dataset suggested by Reviewer Fmsc). As in our previous experiment, we consider SA as the attack, ERM as a baseline (no group robustness method) and report both the ASR and WGA.
> >
> > | Poison %          |    0.5%     |      1%     |
> > | ------------------- | -----------  | ----------- |
> > |AFR (ASR)       |    2.1%       |  14.4%      |
> > |ERM (ASR) |   1.0%      |     5.3%    |
> >
> >
> > | Poison %          |    0.5%      |     1%     |
> > | ------------------- | -----------  | ----------- |
> > |AFR (WGA)        |   75.3%     |      78.4%  |
> > |ERM (WGA)  |  68.0%      |     70.0%   |
> >
> >
> > We observe the same tension in these experiments: while AFR improves the WGA over the baseline (as expected), it also inadvertently amplifies the ASR (e.g., from 5.3% to 14.4%, a significant boost for a weak attacker).
> >
> >
> > [1] Dropout as a Bayesian Approximation: Representing Model Uncertainty in Deep Learning, Gal et al. (2015)
> >
> > [2] The power of ensembles for active learning in image classification, Beluch et al. (CVPR’18)
> >
> > [3] Deep Learning Through the Lens of Example Difficulty, Baldock et al. (NeurIPS’21)
> >
> > [4] Distribution Density, Tails, and Outliers in Machine Learning: Metrics and Applications, Carlini et al. (2019)
> >
> > [5] Poisoning Web-Scale Training Datasets is Practical, Carlini et al. (2023)
> >
> > [6]  Weaponizing Middleboxes for TCP Reflected Amplification, Bock et al. (USENIX Security’21)
> >
> > [7] Simple and Fast Group Robustness by Automatic Feature Reweighting, Qiu et al. (ICML'23)

---

### Official Review · Reviewer_a2i2 · 2023-11-01

**Soundness:** 3 good
**Presentation:** 2 fair
**Contribution:** 3 good
**Rating:** 6
**Confidence:** 3

**Summary:**

This work identifies a critical conundrum between group robustness and poisoning resilience metrics in machine learning. The contributions are 3-fold:

1) It presents empirical evidence illustrating that methods designed for group robustness are unable to differentiate between minority groups and poisoning data. This inability exacerbates the impact of poisoning attacks, a claim further substantiated by theoretical support under various assumptions.

2) Further, it demonstrates through empirical studies that standard defenses against data poisoning struggle to distinguish poisoned data from that of minority groups. This leads to a compromise in group robustness, an issue observed in both centralized and federated learning setups.

3) Finally, this paper concludes that merely combining group robustness strategies with poisoning defense mechanisms fails to address these challenges, indicating a need for more nuanced solutions.

**Strengths:**

1) This paper is well written and it is easy to follow. The takeaways from each section are explicit, novel and clear.

2) Figure 1 clearly conveys the key ideas.

3) The findings are interesting and useful to the community.

**Weaknesses:**

1) Experiments. The results on CelebA (Section 4) only uses a randomly sampled 10% of the dataset which is concerning. Can the authors also consider the full CelebA setup, so that the results are more reliable?

2) Why are the ASR values negative in Table 3 (Waterbirds / SA setup).

3) Could the authors elaborate on whether any experiments were conducted involving the other two types of poisoning defenses mentioned in Section 2.1?



Minor concerns

1) Title Appropriateness: The broad focus implied by the current title, "Poisoning Defenses," does not accurately reflect the specific emphasis of the work on poisons that are challenging to learn. A more precise title would set clearer expectations for readers.



Overall I enjoyed reading this paper. In my opinion, the strengths of this paper outweigh the weaknesses. Authors please consider addressing the weaknesses above during rebuttal.




======================

**Post-rebuttal**

Thank you authors for your detailed response. The rebuttal addresses most of my concerns, and I maintain my initial rating.

**Questions:**

Please see Weaknesses section above for a list of all questions.

---

> ### Author Response · Authors · 2023-11-23
> **Response to Reviewer a2i2**
>
> We thank the reviewer for their constructive feedback! Please find below our response to each of your comments:
>
> > The results on CelebA (Section 4) only uses a randomly sampled 10% of the dataset which is concerning. Can the authors also consider the full CelebA setup, so that the results are more reliable?
>
> To address the reviewer's concern, we ran additional experiments with 20%, 50% and 100% of the CelebA. The results, over 3 runs, showing the WGA in each scenario are in the following table:
>
> |    |20% of CelebA  | 50% of CelebA | Full CelebA      |
> |----|---------------|---------------|---------------|
> |   Ideal  | 52.3 +- 1.6  | 50.3 +- 1.9   | 45.9 +- 5.8 |
> | Standard  | 43.1 +- 5.5  | 46.4 +- 2.8   | 43.6 +- 3.0|
> |   Worst  | 32.3 +- 2.7  | 32.0 +- 0.8   |  37.7 +- 3.8|
>
> The persistent gap between the Ideal and Standard cases demonstrates the negative impact of EPIc on group robustness (i.e., our findings are consistent).
>
>
> > Why are the ASR values negative in Table 3 (Waterbirds / SA setup).
>
> For the SA, we define the ASR as the relative accuracy drop of the attack on the target group over a non-poisoned model. So, in case the defense not only prevents the accuracy drop by removing the poisons but also slightly increases the accuracy on the targeted group, then the ASR would be negative. We will clarify this in the updated paper.
>
>
> > Could the authors elaborate on whether any experiments were conducted involving the other two types of poisoning defenses mentioned in Section 2.1?
>
> Indeed, in our paper, we only consider defenses that assume that poisons are hard to learn as more realistic, weak, attackers are the ones who would benefit from the amplification offered by group robustness methods. However, we believe that poisoning defenses designed against easy-to-learn poisons (e.g., [1]) would indeed not eliminate legitimate minority samples and hence not hurt the worst-group accuracy (though, such defenses struggle against attacks that craft hard-to-learn poisons [2]).
>
> A promising future direction could be the defenses that assume poisons and clean samples follow different distributions, so they create a small, poison-free set of samples for each group to *erase* poisons (similar to [3] but with a group-balanced clean data to preserve group robustness).
>
>
> > The broad focus implied by the current title, "Poisoning Defenses," does not accurately reflect the specific emphasis of the work on poisons that are challenging to learn. A more precise title would set clearer expectations for readers.
>
> We agree with the reviewer's concern that our title might be too broad. We will specify that we study weaker poisoning attacks, e.g., "Resilience to Weak Poisoning Attacks and Group Robustness are at Odds". Please let us know if you have a suggestion!
>
>
> [1] Anti-Backdoor Learning: Training Clean Models on Poisoned Data, Li et al. (NeurIPS'21)
>
> [2] Narcissus: A Practical Clean-Label Backdoor Attack with Limited Information, Zeng et al. (CCS'23)
>
> [3] Neural Attention Distillation: Erasing Backdoor Triggers from Deep Neural Networks, Li et al. (ICLR'21)

---

### Official Review · Reviewer_wCf5 · 2023-11-04

**Soundness:** 2 fair
**Presentation:** 3 good
**Contribution:** 2 fair
**Rating:** 6
**Confidence:** 4

**Summary:**

This paper studies the existing tension between the group robustness and the resilience to poisoning attacks. The authors argued that, group robustness methods will unavoidably amplify the poisoning samples and boost poisoning performance. On the other hand, poisoning defenses will remove poisoning outliers will also unavoidably remove the minority samples. The authors advocate for tacking this inherent tension in future works.

**Strengths:**

1. The problem of the inherent tension between group robustness and poisoning resilience is an interesting problem/
2. The presented empirical results are well-presented to support the main claim in the paper.

**Weaknesses:**

The main weakness of this paper is to make an impossibility type of claim without a more principled understanding on the problem.
In particular, the authors aim to find a tension between any poisoning attack and the distributionally robust optimization methods, which probably undermines the much-needed principled understanding. In particular, I am expecting some type of result that points out the tension between a poisoning with a clearly defined attacker capability and objective, and the group robustness optimization techniques (e.g., using loss-based thresholding). This result should show that, in order to achieve the attacker objective maximally, the poisoning points will also unavoidably become the amplified minority samples. This way, we can make claims on the impossibility of having group robustness and poisoning resilience at the same time. My major concern here is, the tensions between poisoning and group robustness are drawn some limited empirical results and the poisoning attacks also do not reflect what the attacker in practice may really care to do. For example, many ASR are quite low as presented in Table 2 and I am not sure if it is because the poisoning attacks are configured in some undesirable ways (with respect to the attack objective), who may choose a different approach to achieve higher ASR. The alternative, in turn, may no exhibit a strong tension. Related to this, the impossibility result in the paper only shows that there exist some poisoning points that are harder to learn but did not reason whether these poisoned points are useful for achieving the attacker objectives. I would suggest the authors to first clearly define the threat model and then conduct more rigorous analysis on how this can interfere with group robustness methods. Then, the insights are supported through extensive empirical evaluations.

**Questions:**

1. In page 4, what is the intuition behind configuring the poisoning attacks as described? The configuration seems to be different from the one in the original paper and the authors should clearly explain the reason.
2. Will the gradient shaping method [1] enable a mitigation to the currently observed tension?

[1] Hong et al., "On the Effectiveness of Mitigating Data Poisoning Attacks with Gradient Shaping", arXiv 2020.

==========

Thanks for your detailed response. The rebuttal addresses most of my concerns, I have increased my rating.

---

> ### Author Response · Authors · 2023-11-23
> **Response to Reviewer wCf5 (1)**
>
> We thank the reviewer for their constructive feedback! Please find below our response to each of your comments:
>
> > …poisoning attacks also do not reflect what the attacker in practice may really care to do. For example, many ASR are quite low as presented in Table 2 and I am not sure if it is because the poisoning attacks are configured in some undesirable ways…
>
> An attacker in practice aims to achieve the highest ASR possible within their limited budget. In the case of a poisoning adversary, the number of injected samples determines the real-world budget. The lower the budget is, the more realistic an attack becomes (e.g., the attacker needs to compromise fewer websites to inject images [1]).
>
> A main takeaway from our experiments is that low-budget attacks where the adversaries struggle to achieve high ASR (hence the results in Table 2) can benefit from the defender’s use of group robustness methods such as JTT. The same cannot be said for high-budget attacks that can already achieve high ASR (thus, amplification is less critical). Therefore, the low ASR of the attacks we performed (Table 2) is not due to any specific configuration of the attacks (e.g., hyper-parameters) but due to the considered attacker budgets. This allows us to expose how low-budget attacks (more realistic but weaker) can be inadvertently amplified into stronger ones, essentially giving the attacker “more bang for their buck”.
>
> Furthermore, we would like to clarify that for the Subpopulation Attack (SA) in Table 2, ASR indicates a **relative accuracy drop** on the attack's target group over a non-poisoned model. Therefore, some of our experiments already consider more powerful attacks (e.g., when the ASR is 31%), in which our findings are consistent.
>
>
> > I am not sure if it is because the poisoning attacks are configured in some undesirable ways (with respect to the attack objective), who may choose a different approach to achieve higher ASR —----------- In page 4, what is the intuition behind configuring the poisoning attacks as described?
>
> To address the reviewer’s concerns and questions, we considered attackers (DLBD) selecting poison base samples from different groups of CelebA (the results are over 3 runs):
>
> |          | Gr0 (HRG) | Gr1 (HRG)|Gr2 (HRG)|Gr3 (LRG-1)|
> | -------- | --------  | -------- |---------|-----------|
> | Label/Attribute|0/0|0/1|1/0|1/1|
> | ASR      |97.4 +- 0.1|  97.7 +- 0.1 | 91.2 +- 2.3|   0.1 +- 0.1|
>
>
> The maximum ASR holds for Gr1, which is the one we chose in our experiments as a base group for crafting the samples. Since the attacker changes the label of the samples they poison, the label/attribute configuration will become 1/1 which matches the one for LRG-1 (as stated in the paper).
>
> For a dirty-label attack, if the attacker is aware of the groups that are easiest to learn, they can take advantage of it and create poisons from those groups as base samples (this matches the setting from our theoretical result). Also, we show below the average loss of each group when there are no poisons in the dataset:
>
>
> |          | Gr0 (HRG) | Gr1 (HRG)|Gr2 (HRG)|Gr3 (LRG-1)|
> | -------- | --------  | -------- |---------|-----------|
> | Label/Attribute|0/0|0/1|1/0|1/1|
> | Loss Average      |0.16   |       0.07   |       0.79 |          1.90|
>
>
> We observe that the easier a group is to learn (i.e., less average loss), the higher the ASR will be if the attacker crafts a DLBD attack with base samples from that group.

---

> > ### Author Response · Authors · 2023-11-23
> > **Response to Reviewer wCf5 (2)**
> >
> > > I would suggest the authors to first clearly define the threat model …
> >
> > We will clarify the threat model, explaining that the attacker is aware of which groups are the easiest to learn and crafts their poisons accordingly. However, we additionally considered the scenario when the attacker is not aware of the presence of groups, so they just attack one of the two classes of CelebA, with the DLBD. The results are over 3 runs, are in the following table:
> >
> > |          | Base samples in class 0 | Base samples in class 1|
> > | -------- | --------  | -------- |
> > | ASR (Worst)|98.3 +- 0.1 | 97.6 +- 0.3 |
> > | ASR (Standard)      |90.7 +- 1.7    | 97.3 +- 0.5   |
> > | ASR (Ideal)      |0.1 +- 0.0  | 0.5 +- 0.1 |
> >
> >
> > We observe high amplification (i.e., the small gap between the worst and standard cases) no matter which class is targeted by the attacker. As a result, even a less informed attacker, who is unaware of the groups, would benefit from the amplification.
> >
> >
> > > …impossibility result in the paper only shows that there exist some poisoning points that are harder to learn but did not reason whether these poisoned points are useful for achieving the attacker objectives…
> >
> > Let’s consider an ideal targeted poisoning attack where the adversary can simply take the target sample, flip its label into the target class, and inject $N$ copies of this into the training set. The strength of this attack is mainly determined by $N$. Each poisoned point is undeniably useful for achieving this attacker’s objective but the attack can still have a low ASR if $N$ (i.e., the budget) is low enough. Any poisoning attack's success depends on (i) the poison generation algorithm and (ii) $N$. Our work argues that there are combinations of (i) and (ii) where a group robustness method such as JTT will end up amplifying the poisons. Even the strongest, most optimal poisoning attack can be amplified if $N$ is low enough (e.g., when poisons are hard to learn). Our focus is on identifying when this amplification occurs and its implications (e.g., the gap between Worst and Standard cases in Table 2).
> >
> > > Question: Will the gradient shaping method [1] enable a mitigation to the currently observed tension?
> >
> > Gradient shaping relies on a mechanism similar to differential privacy (DP) that injects noise into the training process. DP is already known to hurt the group robustness [2] as noise prevents the model from learning hard-to-learn, minority samples. As a result, gradient shaping can mitigate the poisons we consider in our work (i.e., weaker attacks with low budget) but it’ll also have an undue impact on group robustness that the defender wants to avoid, i.e., the tension still remains, the defender can either have poison resilience and low worst-group accuracy or poison vulnerability and high worst-group accuracy.
> >
> >
> > [1] Poisoning Web-Scale Training Datasets is Practical, Carlini et al. (2023)
> >
> > [2] Differential Privacy Has Disparate Impact on Model Accuracy, Bagdasaryan and Shmatikov (NeurIPS’19)

---

### Author Response · Authors · 2023-11-23
**General Response**

We would like to thank our reviewers for their valuable feedback! We are glad to see that all the reviewers have found our problem interesting and critical for the community. We apologize for posting our responses on the last day, we prioritized finishing the experiments our reviewers suggested. We will also update our paper with these new results, evidence, and discussions to make our paper more comprehensive and compelling.

Below, we broadly summarize the main reviewer concerns and how we addressed them in our responses.

- **Concerns about the smaller datasets used in experiments (Reviewers *a2i2* and *Fmsc*)**: We presented additional experiments on larger or more complex datasets. More precisely, we considered full CelebA (***a2i2*** and ***Fmsc***) and MultiNLI (***Fmsc***) datasets. We will include the results in the Appendix A2 and A3 in an updated version of the paper.

- **Concerns about the group robustness methods considered and the scope of our experiments (Reviewers *Shsu* and *Fmsc*)**: We presented a new discussion on how existing heuristics in many, more recent, group robustness methods have common characteristics to the ones we considered. We also presented new experiments on AFR, a state-of-the-art group robustness method, that supported our claims. We will include the additional citations in the Related Work section and we will add the discussion and the new results in the Appendix A2.

- **Concerns about the threat models or the success of the considered attacks (Reviewers *Shsu* and *wCf5*)**: We clarified our threat model and specified our focus on low-budget, weaker attacks that cannot achieve high success in the standard setting but will benefit from being amplified by group robustness methods.

- **Concerns about limited/unusual configurations of the considered attacks (Reviewers *Shsu* and *wCf5*)**: We ran more experiments with different configurations for the poisoning attacks to show the consistency of our findings. We will include the results in the Appendix A2.

- **Concerns about the baseline models to measure the poison success amplification (Reviewer *Shsu*)**: We presented new results that use a standard ERM model as the baseline, as suggested. We will include the results in the Appendix A2.

---

### Meta-Review · Area_Chair_ijxh · 2023-12-08

**Metareview:**

The paper received mixed reviews. However, in overall the reviewers agreed on the novelty of the main observation of the paper. To that end we propose the accept score for the paper. We would strongly recommend the authors to take deep look at the comments of the reviewers and include the additional experimental results from the rebuttal in the main paper. This will alleviate one of the major concerns of having limited empirical evaluation.

**Justification For Why Not Higher Score:**

The reviews were not strong enough.

**Justification For Why Not Lower Score:**

I could give a lower score, but the fact that the paper provides a novel observation pushed me to accept the paper.

---

### Decision · Program_Chairs · 2024-01-16

Accept (poster)